# ERK-mediated NELF-A phosphorylation promotes transcription elongation of immediate-early genes by releasing promoter-proximal pausing of RNA polymerase II

Seina Ohe[1,2], Yuji Kubota[1], Kiyoshi Yamaguchi[3], Yusuke Takagi[1], Junichiro Nashimoto[1], Hiroko Kozuka-Hata[4], Masaaki Oyama[4], Yoichi Furukawa [3] & Mutsuhiro Takekawa [1,2,4] ✉

Growth factor-induced, ERK-mediated induction of immediate-early genes (IEGs) is crucial for cell growth and tumorigenesis. Although IEG expression is mainly regulated at the level of transcription elongation by RNA polymerase-II (Pol-II) promoter-proximal pausing and its release, the role of ERK in this process remains unknown. Here, we identified negative elongation factor (NELF)-A as an ERK substrate. Upon growth factor stimulation, ERK phosphorylates NELF-A, which dissociates NELF from paused Pol-II at the promoter-proximal regions of IEGs, allowing Pol-II to resume elongation and produce full-length transcripts. Furthermore, we found that in cancer cells, PP2A efficiently dephosphorylates NELF-A, thereby preventing aberrant IEG expression induced by ERK-activating oncogenes. However, when PP2A inhibitor proteins are overexpressed, as is frequently observed in cancers, decreased PP2A activity combined with oncogene-mediated ERK activation conspire to induce NELF-A phosphorylation and IEG upregulation, resulting in tumor progression. Our data delineate previously unexplored roles of ERK and PP2A inhibitor proteins in carcinogenesis.

The ERK pathway transduces mitogenic signals and plays a pivotal role in a wide array of biological processes, including cell proliferation, differentiation, and carcinogenesis[1]. Upon stimulation of cells with growth factors such as epidermal growth factor (EGF), their respective receptor tyrosine kinases (RTKs) activate Ras and recruit Raf family kinases to the plasma membrane, which promotes Raf activation. Activated Raf phosphorylates and activates MEK1/2, which in turn activate ERK1/2 by phosphorylation. A portion of the activated ERK then translocates to the nucleus where it phosphorylates and activates specific substrate proteins, including several transcription factors (TFs) (e.g., ELK1 and Sp1), and promotes the expression of so-called immediate-early genes (IEGs)[2]. IEGs are a class of genes whose

[1]Division of Cell Signaling and Molecular Medicine, Institute of Medical Science, The University of Tokyo, 4-6-1 Shirokanedai, Minato-ku, Tokyo 108-8639, Japan. [2]Department of Computational Biology and Medical Sciences, Graduate School of Frontier Sciences, The University of Tokyo, Chiba 277-8583, Japan. [3]Division of Clinical Genome Research, Institute of Medical Science, The University of Tokyo, 4-6-1 Shirokanedai, Minato-ku, Tokyo 108-8639, Japan. [4]Medical Proteomics Laboratory, Institute of Medical Science, The University of Tokyo, 4-6-1 Shirokanedai, Minato-ku, Tokyo 108-8639, Japan. ✉e-mail: takekawa@ims.u-tokyo.ac.jp

expression is low or undetectable in quiescent cells but is rapidly (within 15–60 min) induced after growth factor stimulation in a manner independent of de novo protein synthesis. Since many of the IEGs encode TFs (e.g., c-Fos and c-Jun), their induction further upregulates the expression of various target genes that elicit cell cycle progression (e.g., cyclin D1), motility, angiogenesis, and others, depending on cell type and stimulus[2,3]. Therefore, some IEGs act as proto-oncogenes and promote tumorigenesis.

Aberrant activation of ERK signaling is frequently observed in human cancers. Gene amplification or gain-of-function mutations of key ERK pathway components, including RTKs (e.g., EGFR), Ras, and BRAF, are common in a variety of cancers[1]. These oncogenes constitutively activate ERK and induce hyperphosphorylation of ERK substrate proteins, each of which contains SP or TP consensus phosphorylation motifs[1,4], thereby leading to carcinogenesis. Moreover, since the phosphorylation level of a protein is determined by the balance between the activities of the related protein kinase and phosphatase, aberrant inactivation of protein phosphatases is also involved in carcinogenesis. Indeed, the activity of protein phosphatase 2A (PP2A), a major cellular Ser/Thr-specific phosphatase, is often suppressed in a variety of cancers by overexpression of endogenous PP2A inhibitor proteins such as SET and cancerous inhibitor of protein phosphatase 2A (CIP2A)[5]. These molecules directly bind to and inhibit PP2A, thereby contributing to tumor development and progression[6]. However, the precise role and target of the PP2A inhibition in cancer are not fully understood.

The transcriptional regulation of gene expression by Pol-II in metazoans is a fundamental but complex process composed of several sequential steps, including initiation, pausing, pause release, elongation, and termination[7]. In general, during the transcription cycle, mRNA synthesis begins with the association of DNA-binding TFs with their target motifs and the recruitment of Pol-II and general TFs to the promoter region of a gene. This initiation stage is the rate-limiting step for gene expression. Recent studies have, however, revealed that the post-initiation steps, in particular Pol-II pausing and its release into productive elongation, also serve as critical checkpoints for gene expression[8]. At the early elongation stage, Pol-II synthesizes short nascent RNAs and then pauses ~20–60 nucleotides downstream from transcription start sites (TSSs) where it awaits further signals to resume elongation and to complete the generation of functional full-length mRNAs. This process, termed promoter-proximal pausing (PPP) of Pol-II, is seen in ~60% of expressed mammalian genes, and its stimulus-induced release into productive elongation is thought to be critical for achieving rapid and synchronous gene expression[9]. Notably, the rapid expression of IEGs is mainly regulated at the level of elongation by PPP and its release[10]. Indeed, the promoter regions of many IEGs (e.g., FOS and JUNB) are constitutively occupied by Pol-II independently of mRNA synthesis in resting cells, and upon growth factor stimulation, Pol-II is released from pausing into active elongation, thereby rapidly inducing IEG expression[11]. However, the precise mechanism by which growth factors elicit Pol-II pause release on IEGs remains unknown.

In general, Pol-II pausing and its release are regulated by the coordinated action of negative and positive regulators, including NELF, DRB sensitivity-inducing factor (DSIF), and positive transcription elongation factor-b (P-TEFb)[8,12]. PPP is induced mainly by the binding of NELF, which is composed of the four subunits NELF-A, -B, -C (or isoform NELF-D that lacks the first nine NELF-C residues) and -E, to the elongation complex that includes Pol-II and DSIF[13]. A recent cryo-electron microscopy study of the paused NELF-Pol-II-DSIF complex showed that, of the four NELF subunits, NELF-A is particularly important for the induction of PPP, as the C-terminal part (residues 189–528) of NELF-A forms a tentacle-like extension that directly binds to Pol-II and DSIF, thereby stabilizing Pol-II pausing[14]. In contrast, Pol-II pause release is mediated by P-TEFb, a kinase complex composed of cyclin-dependent kinase 9 (CDK9) and cyclin T1[15]. P-TEFb phosphorylates Ser[2]

on the C-terminal domain (CTD) of Pol-II as well as DSIF[15,16]. Furthermore, P-TEFb also phosphorylates NELF[17]. These orchestrated phosphorylation events are thought to induce NELF dissociation from paused Pol-II and allow Pol II to resume elongation. However, recent studies indicate that NELF dissociation can occur independently of P-TEFb[18,19]. Indeed, following lipopolysaccharide stimulation, genome-wide NELF dissociation from chromatin takes place normally even in cells genetically deficient for CDK9 (the kinase subunit of P-TEFb)[20], indicating the presence of another as yet unidentified mechanism that controls NELF dissociation. Furthermore, although both ERK signaling and NELF-mediated Pol-II regulation play pivotal roles in the induction of IEGs, their functional relationship, if any, remains unexplored.

Here, we identified NELF-A, the subunit of NELF that is critical for the stabilization of Pol-II pausing, as a substrate of ERK. We also demonstrate the molecular mechanism by which growth factors induce IEG expression at the level of transcription elongation by controlling PPP. Furthermore, we show that NELF-A phosphorylation is tightly regulated by the balance between ERK and PP2A activities. Therefore, in cancer cells with intact PP2A, oncogene-induced constitutive, but relatively moderate, ERK activity often fails to induce NELF-A phosphorylation, and thus IEG expression is low. However, when PP2A inhibitor proteins, such as SET and CIP2A, are overexpressed, as is frequently found in human cancers, decreased PP2A activity allows enhanced NELF-A phosphorylation; this causes aberrant upregulation of IEGs and associated downstream growth-promoting genes, and tumor progression. Our data provide insights into the roles of ERK and PP2A inhibitor proteins in the etiology of human cancer.

## Results
### Identification of NELF-A as a substrate of ERK

To identify ERK substrates, we screened human cDNA libraries using a yeast three-hybrid method modified from our previous study[4] (Fig. 1a). In this system, the bait was composed of the LexA DNA-binding domain fused to the WW domain of Pin1, which specifically binds to phosphorylated SP or TP sites. The yeast reporter strain carrying the bait plasmid was transformed with a library of prey plasmids in which human cDNA sequences were fused to the GAL4 activation domain, together with a third plasmid that expresses a constitutively active human ERK(PD) mutant. Therefore, in principle, the bait can interact with the prey only when a prey protein derived from human cDNA libraries is phosphorylated by ERK(PD) in the yeast cells, which allows the transfected yeast cells to grow on a selective medium lacking histidine. Out of $1.29 \times 10^6$ human cDNA clones derived from liver, brain, and placenta, we obtained 11 His-positive clones, one of which encoded NELF-A, a component of the NELF complex. A WW domain mutant that had lost its ability to bind to phosphorylated SP/TP motifs, i.e., WW(W34A)[21], failed to interact with NELF-A, confirming phosphorylation-dependent interaction (Fig. 1b).

To test directly whether ERK can phosphorylate NELF-A, an in vitro kinase assay was performed. We purified active HA-ERK2 from HEK293 cells co-expressing constitutively active MEK1(DD) (an ERK activator), and incubated it with recombinant GST-NELF-A. The reaction mixture was then separated on SDS-PAGE, and the phosphorylation of GST-NELF-A was monitored by Pro-Q diamond gel staining, which selectively detects phosphorylated proteins (Fig. 1c). Active ERK2, but not a kinase-defective K52N mutant [ERK2(K/N)], directly phosphorylated GST-NELF-A in vitro. Next, to examine whether ERK phosphorylates NELF-A in human cells, we employed an anti-phospho-SP/TP monoclonal antibody. When Flag-NELF-A was coexpressed with MEK1(DD) in HEK293 cells, the antibody reacted strongly with immuno-purified Flag-NELF-A (Fig. 1d). Similarly, EGF stimulation of HEK293 cells expressing Flag-NELF-A activated ERK, and concomitantly induced Flag-NELF-A phosphorylation (Fig. 1e). This phosphorylation was abolished when ERK activity was blocked using the MEK inhibitor, trametinib, indicating ERK-dependent phosphorylation.

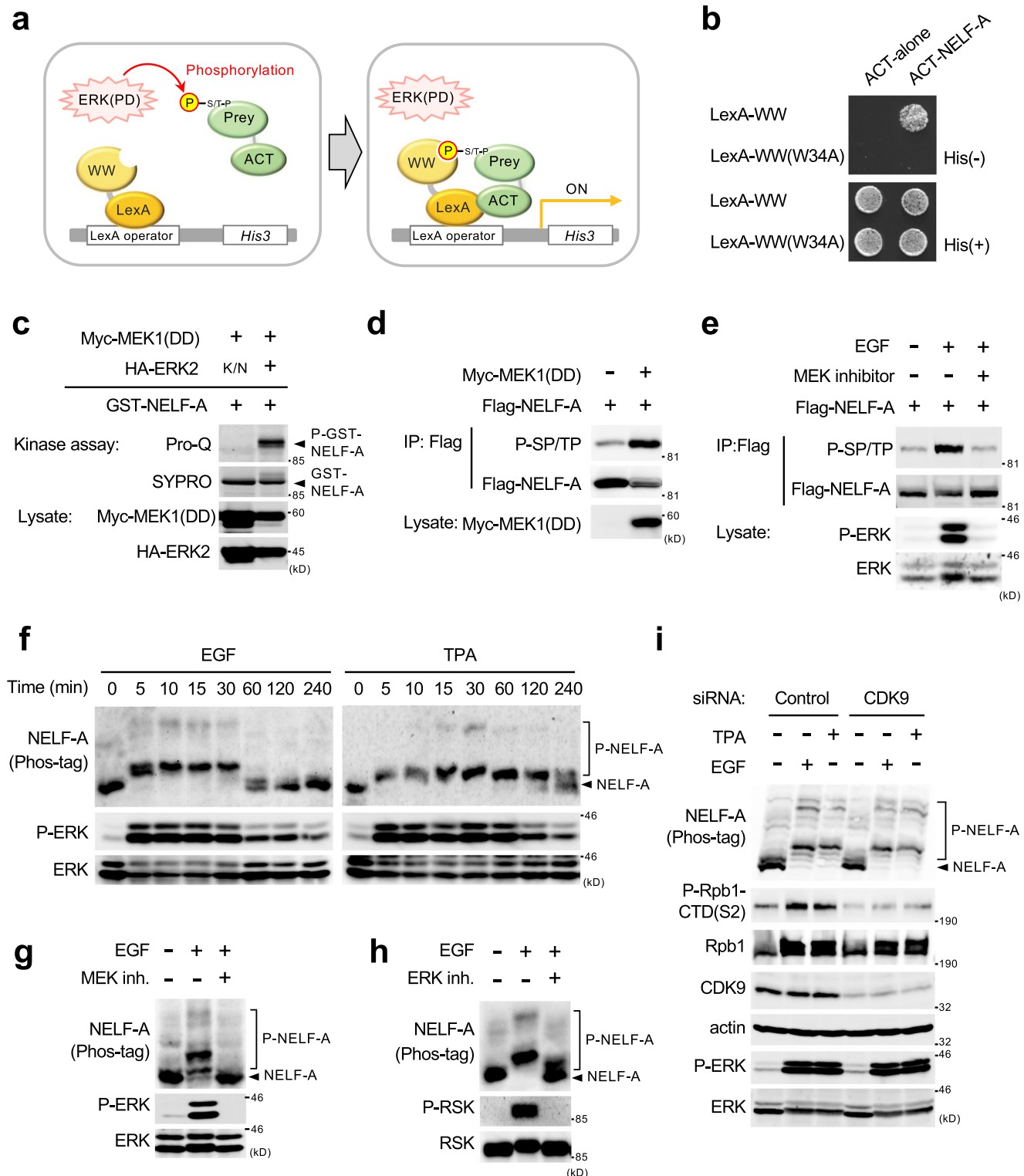

To further confirm endogenous NELF-A phosphorylation in vivo, we applied the Phos-tag SDS-PAGE technique, which leads to retarded electrophoretic mobility of phosphorylated proteins and thus visualizes them as up-shifted bands[22]. HeLa cells were stimulated with EGF, and the cell lysates were separated on Phos-tag SDS-PAGE and immunoblotted for endogenous NELF-A. Following EGF treatment, while conventional SDS-PAGE showed no apparent band-shift (Supplementary Fig. 1a), the Phos-Tag SDS-PAGE exhibited up-shifted (i.e., phosphorylated) NELF-A bands, concomitantly with ERK activation (Fig. 1f, left, and Supplementary Fig. 1b). EGF stimulation rapidly (<5 min)

induced NELF-A phosphorylation, which remained high for at least 30 min before gradual dephosphorylation up to 120 min. Likewise, treatment of HEK293 cells with the ERK activator, tetradecanoyl-phorbol acetate (TPA), leads to similar upward band-shifts of NELF-A (Fig. 1f, right). Such band shifts were abolished in the presence of trametinib or an ERK inhibitor (FR180204) in HeLa cells (Fig. 1g, h) and HEK293 (Supplementary Fig. 1c). In contrast, siRNA-mediated depletion of CDK9 (the kinase subunit of P-TEFb) did not inhibit the NELF-A phosphorylation induced by EGF or TPA, although it substantially suppressed the Pol II-CTD phosphorylation at Ser[2], confirming P-TEFb-

**Fig. 1 | NELF-A is a substrate of ERK. a** Yeast three-hybrid system. WW, the trp-trp domain of Pin1; LexA, a DNA-binding domain; ACT, activation domain. LexA-WW binds prey peptides that are phosphorylated by ERK(PD), thereby inducing *HIS3* reporter expression. **b** Reporter yeast cells expressing ERK(PD) and LexA-WW (wild-type or its non-phospho-SP/TP-binding W34A mutant) were transformed with ACT-NELF-A or empty vector (ACT-alone). Cells were spotted on selective plates with (+) or without (−) histidine. **c** ERK phosphorylates NELF-A in vitro. Purified recombinant GST-NELF-A was incubated with immunopurified HA-ERK2 in vitro, and GST-NELF-A phosphorylation was detected by ProQ diamond gel staining (top). Total GST-NELF-A was visualized by SYPRO Ruby staining (second). Myc-MEK1(DD) and HA-ERK2 expression levels are shown in the lower panels. MEK1(DD), a constitutively active MEK1(S218D/S222D) mutant; K/N, a kinase-inactive ERK2(K52N) mutant. **d** HEK293 cells were co-transfected with Flag-NELF-A and Myc-MEK1(DD). Phosphorylation of immunoprecipitated Flag-NELF-A was probed with anti-phospho-SP/TP antibody (top). **e** Flag-NELF-A-transfected cells were pretreated with (+) or without (−) a MEK inhibitor trametinib (10 μM) and stimulated with EGF (20 ng/ml, for 15 min). The phosphorylation status of immunoprecipitated Flag-NELF-A was assessed as in **d**. **f** HeLa (left) and HEK293 (right) cells were stimulated with EGF or TPA (300 nM) for the indicated times. Lysates were separated on Phos-tag SDS-PAGE and immunoblotted with anti-NELF-A antibody. P-NELF-A, phosphorylated NELF-A. **g**, **h** HeLa cells were pretreated with (+) or without (−) a MEK inhibitor (trametinib, 10 μM) (**g**) or an ERK inhibitor (FR180204, 40 μM) (**h**) and stimulated with EGF (20 ng/ml, for 15 min) as indicated. Phosphorylation status of endogenous NELF-A was analyzed as in **f**. **i** HeLa cells were transfected with control siRNA or siRNA targeting CDK9. One day after transfection, the cells were stimulated with EGF or TPA for 15 min, and the phosphorylation states of NELF-A (top) and Ser2 in the CTD of Pol-II (Rpb1) (second) were assessed by Phos-tag SDS-PAGE and immunoblotting. Actin, loading control. **e**–**i** Cell lysates were probed for phosphorylated ERK1/2 (P-ERK), ERK1/2, phosphorylated RSK (an ERK substrate), and RSK by immunoblotting (lower rows) as indicated. Source data are provided as a Source Data file.

---

independent phosphorylation of NELF-A (Fig. 1i). Virtually identical results were obtained when cellular CDK9 activity was inhibited by its specific inhibitor DRB[15] (Supplementary Fig. 1d). We conclude from the combined data that NELF-A is a bona fide substrate of ERK and is rapidly phosphorylated in vivo in response to mitogenic stimuli through the ERK pathway.

## ERK phosphorylates NELF-A at evolutionarily conserved Ser/Thr residues

Human NELF-A possesses 17 SP/TP sites, of which 16 motifs are conserved in vertebrates. To identify ERK phosphorylation site(s) in NELF-A, we performed mass spectrometry analysis of Flag-NELF-A purified from TPA-stimulated HEK293 cells. In the LC-MS/MS analysis, which achieved about 76% sequence coverage of NELF-A, we identified one phosphorylation site, $S^{363}P$ (Fig. 2a and Supplementary Fig. 2a). Ser-to-Ala substitution at this site (S363A) indeed abrogated some, but not all, of the NELF-A phosphorylation (Fig. 2b). This S363A mutant was, however, still robustly phosphorylated in response to TPA stimulation in cells, as monitored by Pro-Q diamond staining (Fig. 2c), suggesting the presence of other phosphorylation sites. We therefore introduced additional Ala substitutions into the SP/TP sites in the C-terminal SP/TP-rich region (residues 382-400) of NELF-A, as this region was not covered by our mass spectrometry analysis. A combination of the S363A mutation and Ala substitutions of the highly conserved three SP/TP sites ($S^{393}$, $T^{396}$, and $T^{399}$) within the SP/TP-rich region (hereafter termed 4A mutations) resulted in a dramatic decrease in TPA-induced NELF-A phosphorylation (Fig. 2a–c), whereas combinations of S363A and any one of the single mutations at these three SP/TP sites did not (Supplementary Fig. 2b). Furthermore, 7A mutations (4A plus S382A/T385A/T387A) and 16A mutations (in which all the conserved SP/TP sites in NELF-A are mutated to Ala) greatly reduced NELF-A phosphorylation, but at a level similar to that observed in 4 A mutations (Fig. 2c), indicating that these four residues are the major phosphorylation sites of NELF-A. Corroborating these results, the NELF-A(4A) mutant failed to interact with LexA-WW in the yeast three-hybrid assay (Fig. 2d). Therefore, ERK phosphorylates NELF-A at the conserved $S^{363}$, $S^{393}$, $T^{396}$, and $T^{399}$ residues that are located within its tentacle region critical for Pol-II binding[14]. Finally, we investigated whether other NELF subunits could also be targets for ERK-mediated phosphorylation. Pro-Q diamond staining and Phos-tag SDS-PAGE analyses showed that, unlike NELF-A, TPA stimulation did not markedly affect the phosphorylation states of the other NELF subunits (Fig. 2e). Thus, NELF-A most likely serves as the specific substrate of ERK in the NELF complex.

## ERK-mediated NELF-A phosphorylation promotes transcriptional elongation of IEGs

Although both NELF and ERK are key regulators of growth factor-induced gene expression, their functional relationship is unknown.

Since NELF controls PPP of Pol II and its release, we next investigated if ERK-mediated NELF-A phosphorylation plays a role in this process. For this purpose, we established HEK293 cells in which endogenous NELF-A was depleted by shRNA and replaced with shRNA-resistant Myc-NELF-A (wild-type) or the unphosphorylatable Myc-NELF-A(4A) mutant (hereafter called HEK293-WT and HEK293-4A cells, respectively) (Fig. 3a and Supplementary Fig. 3a). As anticipated, EGF-induced NELF-A phosphorylation was profoundly suppressed in HEK293-4A cells as compared with HEK293-WT cells, even though comparable levels of ERK activation and ELK1 phosphorylation (an ERK substrate TF) were observed in both cell lines (Fig. 3b, c). Furthermore, there were no marked differences between these two cell lines in subcellular localization of NELF-A (Fig. 3d), formation of the NELF complex, and phosphorylation states of the Pol II-CTD at $Ser^2$ and $Ser^5$ (Supplementary Fig. 3b, c). We then performed genome-wide transcriptome analyses using Ion AmpliSeq RNA sequencing technology. Total RNA was extracted from these two cell lines before and 60 min after EGF stimulation and was used for the analysis of their gene expression profiles. Based on three independent experiments in each cell line, we observed that EGF stimulation significantly upregulated a total of 210 and 191 genes (adjusted $p < 0.05$, >1.5-fold increase vs. no stimulation) in HEK293-WT and −4A cells, respectively (Supplementary Fig. 3d, e). Comprehensive analysis of these EGF-inducible genes revealed that the induction of a group of genes was significantly suppressed in HEK293-4A cells vs. WT cells (Fig. 3e). Heatmap analysis showed that many such genes were known IEGs[23] (Fig. 3f). We validated the expression levels of the representative IEGs (*GADD45B*, *JUNB*, *FOS*, *FOSB*, and *EGR2*) using quantitative reverse transcription-PCR (qRT-PCR) (Fig. 3g). Interestingly, growth factor-mediated induction of these genes is regulated in part by PPP of Pol-II and its stimulus-dependent release[24,25], implying functional relevance of the NELF-A phosphorylation in PPP regulation. The decreased expression of c-Fos and GADD45β in EGF-treated HEK293-4A cells was also confirmed at the protein level by western blot analyses (Fig. 3c). In contrast to the IEGs, no significant differences were observed between these two cell lines with regard to expression of a housekeeping gene (*TBP*) or of stress-inducible genes (*HSP90AB1* and *HSP90AA1*) whose induction is regulated by PPP under stress conditions[17] (Fig. 3g), suggesting that NELF-A(4A)-mediated transcriptional repression occurs in a gene- and stimulus-specific manner. Thus, these findings indicate that ERK-mediated NELF-A phosphorylation is crucial for growth factor-induced robust expression of the IEGs.

After initiating transcription of target genes, NELF rapidly binds to and pauses Pol-II near their TSSs, and, upon stimulation, it is dissociated from the stalled Pol-II, which allows Pol-II to resume productive elongation and thus to move away from the TSSs to gene-body regions[8]. To determine whether ERK-mediated NELF-A phosphorylation impacts on these elongation steps, we next analyzed the association of NELF-A and Pol-II with two representative IEG loci

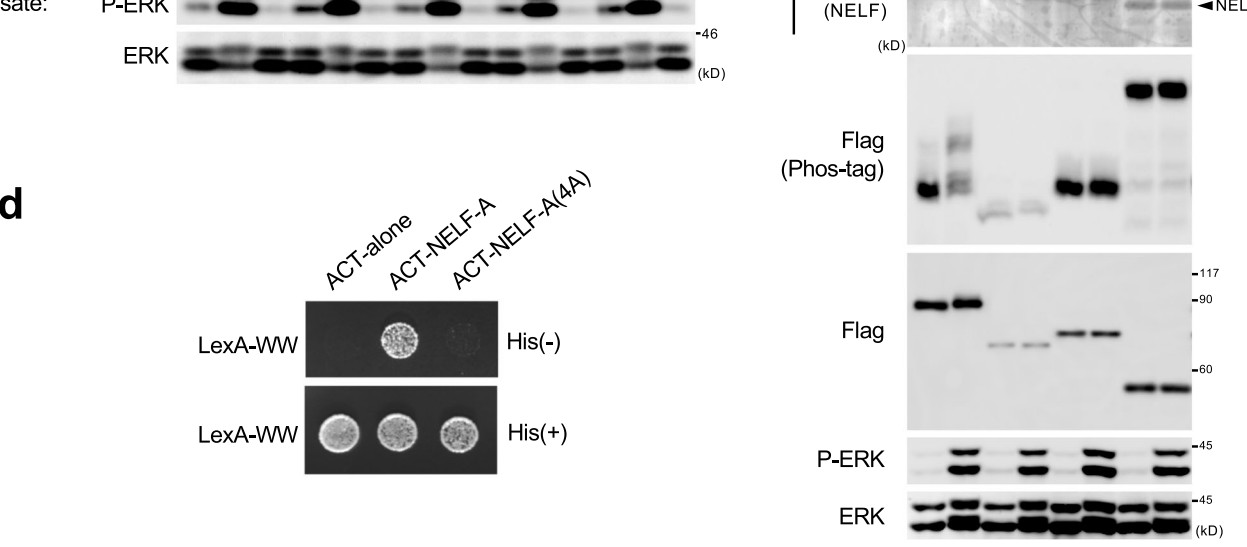

**Fig. 2 | ERK phosphorylates NELF-A at evolutionarily conserved Ser and Thr residues. a** Sequence alignments of various NELF-A orthologues. Conserved residues are shown in blue. Green boxes, ERK phosphorylation sites (SP and TP). The Ser/Thr residues in which Ala substitution mutations were made (S363A, 4A, and 7A) are indicated. **b** HEK293 cells were cotransfected with His-NELF-A (WT, S363A, or 4A) and Myc-MEK1(DD), and treated with (+) or without (−) the MEK inhibitor (trametinib). Phosphorylation of His-NELF-A was analyzed by Phos-tag SDS-PAGE and immunoblotting using an anti-His antibody (upper). The expression level of Myc-MEK1(DD) in the cell lysates is also shown (lower). **c** HEK293 cells expressing Myc-NELF-A or its mutant derivatives (S363A, 4A, 7A, or 16A) were pretreated with or without a MEK inhibitor (AZD6244, 10 μM) and stimulated with TPA (300 nM, for 15 min). The phosphorylation status of immunoprecipitated Myc-NELF-A was assessed by ProQ diamond gel staining (top), and total Myc-NELF-A was

subsequently monitored by SYPRO Ruby staining (second). **d** The reporter yeast cells expressing ERK(PD) and LexA-WW were transformed with ACT-NELF-A, ACT-NELF-A(4A), or the empty vector (ACT-alone), and were then spotted on selective plates with (+) or without (−) histidine. **e** HEK293 cells expressing Flag-NELF-A, -B, -C, or -E were stimulated with TPA for 15 min. Phosphorylation states of immunoprecipitated Flag-NELF subunits were detected by Pro-Q diamond gel staining (top), and their total protein levels were monitored by silver staining (second). The cell lysates were also separated by Phos-tag SDS-PAGE (third) or by conventional SDS-PAGE (fourth), and were immunoblotted with an anti-Flag antibody for phosphorylation states and total levels of the Flag-NELF subunits, respectively. **c, e** Cell lysates were probed for phosphorylated ERK1/2 (P-ERK) and ERK1/2 by immunoblotting (lower rows). Source data are provided as a Source Data file.

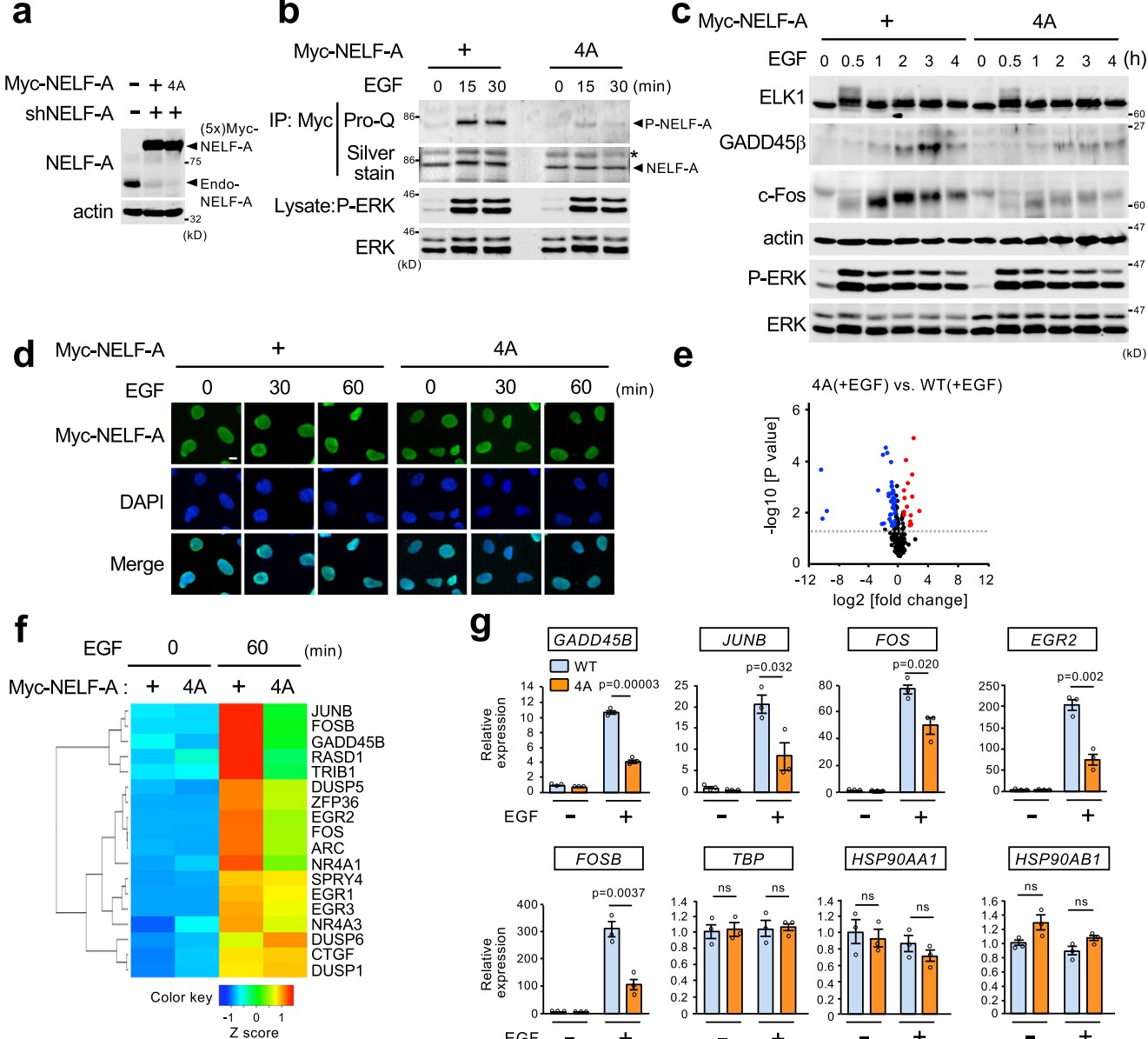

**Fig. 3 | ERK-mediated NELF-A phosphorylation promotes the expression of IEGs. a** HEK293 cells stably expressing shRNA-resistant Myc-NELF-A (WT or 4A) were depleted of endogenous NELF-A using shRNA. Levels of the Myc-NELF-A proteins were confirmed by immunoblotting. Actin, loading control. **b–d** HEK293-WT and −4A cells were stimulated with EGF (20 ng/ml). In (**b**), Myc-NELF-A or Myc-NELF-A(4A) immunoprecipitates were assessed for phosphorylation using Pro-Q diamond gel staining (top). Total Myc-NELF-A was visualized with silver-stain (second). In (**c**), ELK1, GADD45β, and c-Fos levels were monitored by immunoblotting. Up-shifted bands of ELK1 indicate phosphorylation. Actin, loading control. In (**b**, **c**), lysates were immunoblotted for phospho- and total ERK1/2 (lower panels). In (**d**), subcellular localization of Myc-NELF-A or Myc-NELF-A(4A) was detected by immunofluorescent staining with anti-Myc antibody (green). DAPI, nuclear staining.

Scale bar, 10 μm. **e**, **f** HEK293-WT and −4A cells were stimulated with or without EGF for 60 min. **e** Volcano plot comparing expression of the 210 EGF-inducible mRNAs in HEK293-WT and −4A cells (see Supplementary Fig. 3d). Each blue dot represents a down-regulated (FC < 0.66, p < 0.05) mRNAs (46 genes) and each red dot represents an upregulated (FC > 1.5, p < 0.05) mRNAs (15 genes) in HEK293−4A relative to HEK293-WT cells. **f** Heatmap of IEGs comparing HEK293-WT and −4A cells before and 60 min after EGF stimulation. Colors represent high (red), low (blue), or average (green) expression values based on Z-score. **g** HEK293-WT and −4A cells were stimulated with or without EGF for 60 min. The indicated mRNAs were quantified using qRT-PCR. All data were normalized to the level of *GAPDH* expression. *P*-values were assessed using a two-tailed Student's *t*-test. Error bars, SEM (*n* = 3). Source data are provided as a Source Data file.

(*FOS* and *JUNB*) before and 30 min after EGF stimulation by chromatin immunoprecipitation-qPCR (ChIP-qPCR) assays with primer sets specific to the TSSs and other regions of the genes (Fig. 4a, c). In HEK293-WT cells (blue bars in Fig. 4b and Supplementary Fig. 3f), consistent with previous reports related to PPP[11,25], NELF-A was enriched at the TSS of the *FOS* gene under basal conditions (without EGF), and the NELF-A occupancy remained high after EGF stimulation (because EGF increases the transcriptional rate of the *FOS* gene and thus facilitates not only release but also recruitment of NELF at the TSS). Furthermore, in the

absence of EGF, Pol-II was associated only with the TSS of the *FOS* gene, but not with its downstream gene body regions, indicating that it was proximally paused. However, following EGF stimulation, Pol-II accumulated in the gene body regions, confirming EGF-induced transition of paused Pol-II into productive elongation in HEK293-WT cells. By contrast, in HEK293-4A cells (shown in orange bars), significantly more NELF-A(4A) was found at TSS, compared with WT NELF-A in HEK293-WT cells. In particular, the enhanced association of NELF-A(4A) with the TSS became more evident following EGF treatment, suggesting that

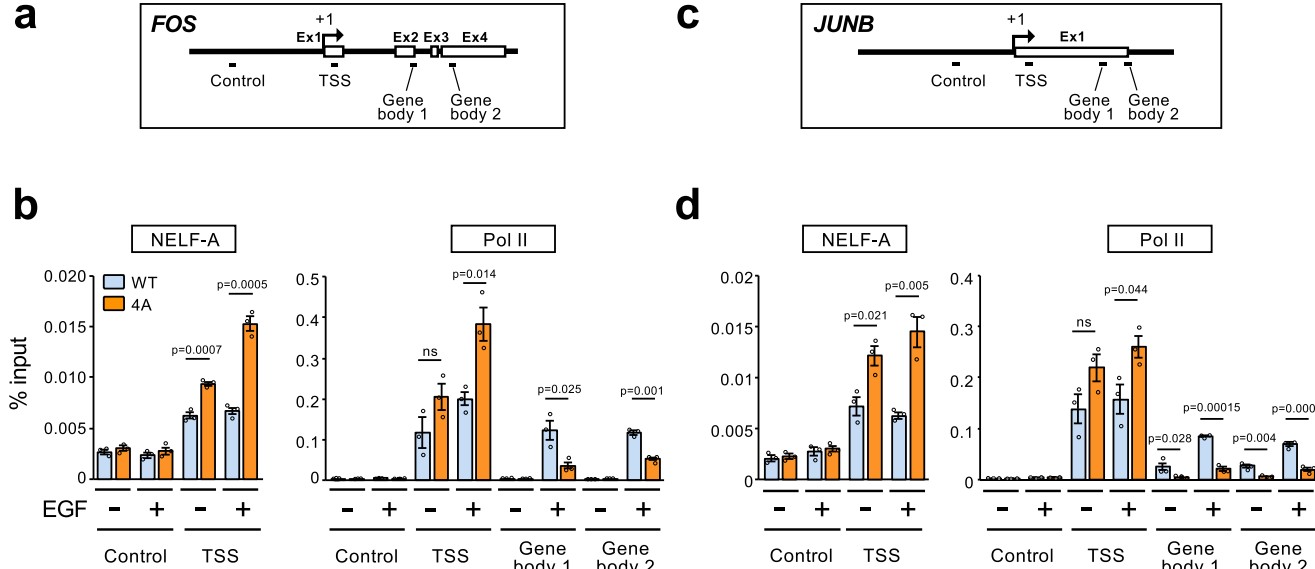

**Fig. 4 | ERK-mediated NELF-A phosphorylation promotes transcriptional elongation of IEGs. a, c** Schematic diagrams of the *FOS* (**a**) and *JUNB* (**c**) genomic loci. Boxes represent exons and bars below the genes show the positions of amplicons used for ChIP-qPCR analyses. TSS, transcription start site. **b, d** Distribution of NELF-A and RNA polymerase II (Pol-II) on the *FOS* (**b**) and *JUNB* (**d**) genes before and after EGF stimulation. ChIP-qPCR assays were performed with anti-NELF-A antibody (left) or anti-Pol-II antibody (right) using chromatin prepared from HEK293-WT or −4A cells that were simulated with (+) or without (−) EGF (20 ng/ml for 30 min). The densities of NELF-A and Pol-II are presented as % of input. *P*-values were assessed using a two-tailed Student's *t*-test. Error bars, SEM (*n* = 3). Source data are provided as a Source Data file.

NELF-A(4A) efficiently binds to and stalls Pol-II at the promoter-proximal site even after EGF stimulation. Indeed, after EGF addition, the amount of paused Pol-II near the TSS was significantly increased, whereas the levels of processive Pol-II on the gene body regions were markedly decreased in HEK293-4A cells as compared with HEK293-WT cells. Thus, Pol-II remained stalled at the TSS even after EGF stimulation when cellular NELF-A phosphorylation is prevented. Essentially identical results were also obtained for *JUNB* (Fig. 4d and Supplementary Fig. 3g). These results indicate that ERK-mediated phosphorylation of NELF-A is crucial for growth factor-induced eviction of NELF from paused Pol-II and consequent productive Pol-II elongation, and thus promotes IEG expression.

## ERK-mediated NELF-A phosphorylation promotes cell growth and tumorigenesis

Since many IEGs encode TFs that induce growth-promoting genes, including cyclin D1[3], ERK-mediated NELF-A phosphorylation and the resulting robust expression of IEGs likely contributes to cell cycle progression and proliferation. To test this possibility, we utilized immortalized human epidermal keratinocytes (HaCaT) carrying intact retinoblastoma (Rb) protein, a key regulator of growth factor-induced cell cycle progression[26]. We again established HaCaT cell lines in which endogenous NELF-A is depleted and replaced with the Myc-NELF-A or Myc-NELF-A(4A) (termed HaCaT-WT and HaCaT-4A cells, respectively) (Fig. 5a). These cells were synchronized in the G1 phase by serum-starvation and then released into the cell cycle by EGF addition. In agreement with the results obtained in HEK293, EGF-induced NELF-A phosphorylation (Fig. 5b) and the expression of the representative IEGs were significantly repressed in HaCaT-4A cells as compared with HaCaT-WT cells (Fig. 5c). Furthermore (and consistent with the decreased IEG expression), EGF-induced cyclin D1 protein expression (Fig. 5d), the rate of bromodeoxyuridine (BrdU) incorporation (a marker for S-phase entry) (Fig. 5e), and consequent cell proliferation (Fig. 5f) were all significantly attenuated in HaCaT-4A cells relative to WT control cells. Thus, NELF-A phosphorylation and the resulting efficient transcriptional elongation of IEGs are crucial for growth factor-induced cell cycle progression and proliferation.

To further validate these findings in vivo, we employed a mouse xenograft model. Although HaCaT cells do not develop into invasive cancers, they can form nodular tumors when inoculated into athymic nude mice[27,28]. We subcutaneously injected HaCaT-WT or −4A cells into nude mice and monitored tumor growth for 36 days (Fig. 5g). The tumors formed by HaCaT-4A cells were significantly smaller than those derived from HaCaT-WT cells. Corresponding changes in tumor weight were also observed (Fig. 5h). Moreover, similar results were observed when HaCaT-WT and −4A cells transformed with H-Ras(G12V) (termed Ras-HaCaT-WT and Ras-HaCaT-4A cells, respectively) were transplanted into nude mice (Supplementary Fig. 4a–c). These cell lines developed aggressive tumors in vivo. However, the tumor volume and weight of mice injected with Ras-HaCaT-4A cells were significantly lower than those of mice injected with Ras-HaCaT-WT cells (Fig. 5i, j). Thus, ERK-mediated NELF-A phosphorylation promotes tumor growth in vivo by facilitating the expression of IEGs as well as their downstream growth-promoting genes.

## PP2A counteracts oncogene-induced NELF-A phosphorylation in cancer cells

ERK signaling is frequently hyper-activated by various oncogenes in human cancers. Therefore, on the basis of the above results, we postulated that NELF-A would be persistently phosphorylated in such cancer cells. To test this prediction, we assessed the phosphorylation status of NELF-A in three representative cancer cell lines harboring various ERK-activating oncogenes: A431 (EGFR amplification), H1299 (NRas^Q61K), and A375 (BRaf^V600E). In control cells (HeLa) with low basal ERK activity, EGF-induced strong ERK activation led to a marked phosphorylation of endogenous NELF-A. Surprisingly, however, little or no phosphorylation of NELF-A was detected in the three cancer cell lines, even though ERK and RSK (an ERK substrate) were constitutively phosphorylated and activated by the oncogenes (Fig. 6a). To address this unexpected result, we considered the possibility that a protein phosphatase counteracts NELF-A phosphorylation. Initially, to identify the protein phosphatase responsible for NELF-A dephosphorylation in vivo, HeLa cells were stimulated with TPA in the presence of various phosphatase inhibitors and the phosphorylation-dephosphorylation

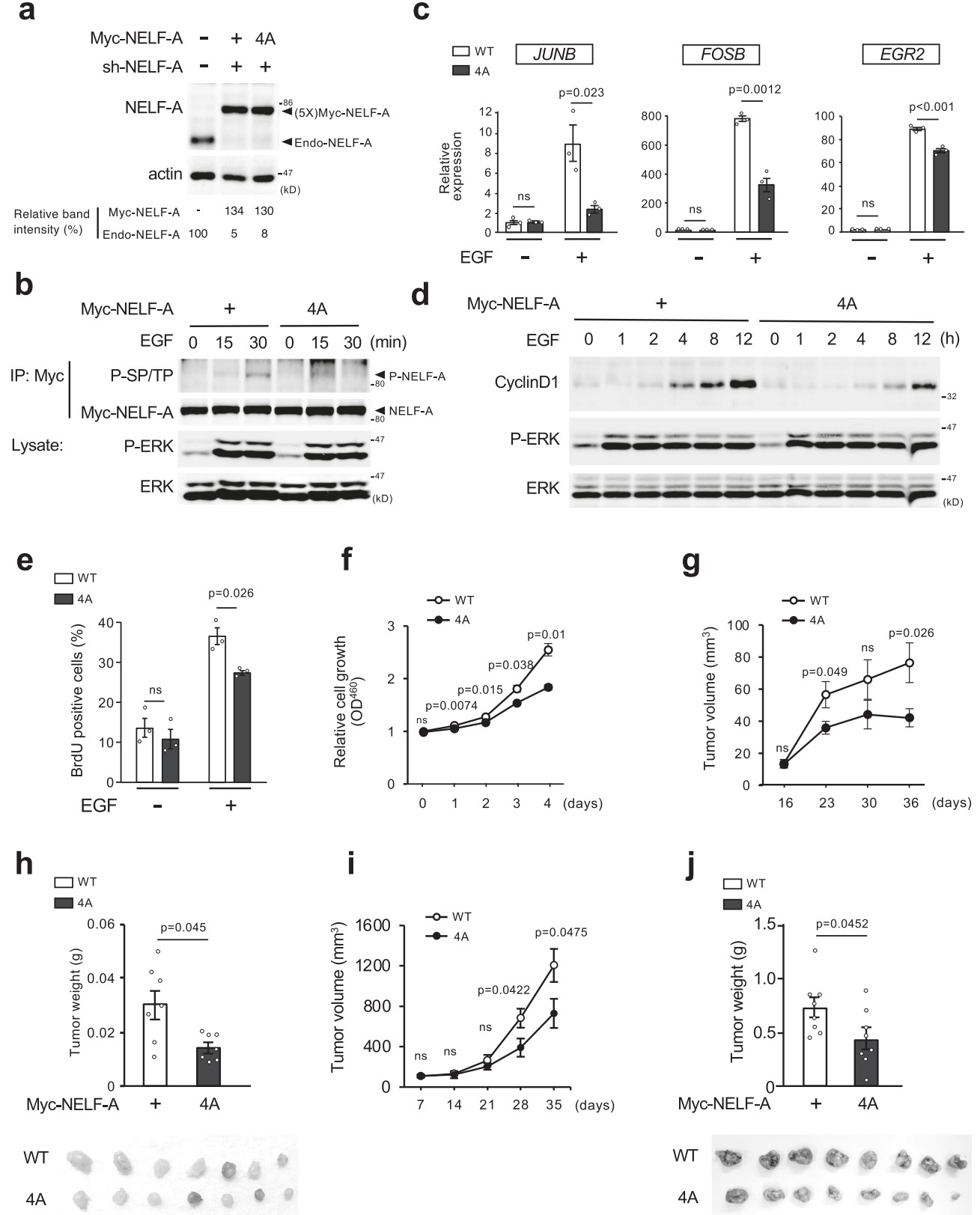

kinetics of NELF-A was monitored by Phos-tag SDS-PAGE (Fig. 6b–d and Supplementary Fig. 5a, b). Treatment of the cells with a PP2A inhibitor, okadaic acid (OA), largely suppressed the dephosphorylation of NELF-A and thus retained NELF-A in the phosphorylated forms for an extended period of time, whereas sodium orthovanadate (an inhibitor for dual-specificity MAPK phosphatases and tyrosine phosphatases) did not

(Fig. 6b). Similarly, two highly specific PP2A inhibitors[29], cytostatin (Fig. 6c) and rubratoxin A (Fig. 6d), but not a PP2B inhibitor (FK506) or a PP2C inhibitor (Sanguinarine)[30], suppressed NELF-A dephosphorylation. Furthermore, when NELF-A was coexpressed with one of two different PP2A inhibitor proteins, SET or CIP2A, in HEK293 cells, substantial increases in NELF-A phosphorylation were observed

**Fig. 5 | ERK-mediated NELF-A phosphorylation promotes cell growth and tumorigenesis. a** HaCaT cells stably expressing shRNA-resistant Myc-NELF-A (WT or 4A) were depleted of endogenous NELF-A with shRNA. Levels of Myc-NELF-A proteins were confirmed by immunoblotting. Actin, loading control. **b, d** HaCaT-WT and −4A cells were synchronized at the G1-phase by serum starvation, and stimulated with EGF for the indicated times. In (**b**), phosphorylation of immuno-precipitated Myc-NELF-A (WT and 4A) was probed with anti-phospho-SP/TP anti-body. In (**d**), cell lysates were probed for cyclin D1, phosphorylated ERK1/2 (P-ERK) and ERK1/2 by immunoblotting using the appropriate antibodies. **c** HaCaT-WT and -4A cells were stimulated with or without EGF (60 ng/ml, for 60 min). Total RNA was extracted and analyzed for the indicated mRNA expression using qRT-PCR. All data were normalized to the level of *GAPDH* expression. **e** Serum-starved HaCaT-WT and

-4A cells were stimulated with EGF for 18 h, and BrdU- positive nuclei were quantified on the Operetta imaging system. More than 5000 cells were counted in each experiment. **f** The indicated cells were cultured in medium with 10% FBS for 4 days. Cell proliferation was determined using a CCK8 assay. OD, optical density. **c, e, f**, Error bars, SEM (*n* = 3). **g–j** HaCaT-WT and -4A cells (**g, h**) or Ras-HaCaT-WT and -4A cells (**i, j**) were subcutaneously injected into BALB/c nude mice. In (**g, i**), the tumor volume was monitored on the indicated days. In (**h, j**), the weight of excised xenograft tumors on day 36 (**h**) and on day 35 (**j**) (upper). Photographs of the xenograft tumors excised from individual mice (lower). Error bars, SEM (*n* = 7 in (**g**) and (**h**), *n* = 8 in **i** and **j**). **c, e–j**, *P*-values were assessed using a two-tailed Student's t-test. Source data are provided as a Source Data file.

(Fig. 6e, f). Conversely, treatment with a PP2A activator, FTY720[31], accelerated NELF-A dephosphorylation (Fig. 6g). Finally, we confirmed that recombinant PP2A directly dephosphorylated NELF-A in vitro (Fig. 6h). Based on these combined findings, we conclude that PP2A is the major phosphatase responsible for NELF-A dephosphorylation in cells. We then determined whether PP2A indeed counteracted the oncogene-induced NELF-A phosphorylation in cancer cells with ERK-activating oncogenes. Interestingly, inhibition of cellular PP2A activity by OA sufficed to induce strong NELF-A phosphorylation in various cancer cell lines (H1299, A375, A431, and A549) even though a comparable level of ERK activity was observed before and after OA treatment in each cell line (Fig. 6i). This OA-induced NELF-A phosphorylation was abolished in the presence of the MEK inhibitor, trametinib, in H1299 cells, indicating the requirement of ERK activity for this phosphorylation (Fig. 6j). Thus, these findings indicate that NELF-A phosphorylation status in vivo is tightly regulated by a balance between ERK and PP2A activities, and that PP2A can overwhelm ERK to settle NELF-A in an unphosphorylated form in these cancer cells even though basal ERK activity is elevated by various oncogenes.

Next, to further determine the role of PP2A in NELF-A-dependent IEG expression, we treated the HEK293-WT and −4A cells with EGF, with or without OA, and quantified the expression levels of *JUNB* and *FOSB* (Fig. 7a and Supplementary Fig. 5c). In HEK293-WT cells, EGF-induced *JUNB* expression was markedly enhanced in the presence of OA (Fig. 7a, left). Importantly, this OA-mediated potentiation of *JUNB* induction was dramatically diminished in HEK293-4A cells that express the unphosphorylatable NELF-A(4A). Similar results were obtained for *FOSB* expression (Fig. 7a, right). Furthermore, we confirmed that the highly specific PP2A inhibitors, rubratoxin A and cytostatin, produced virtually identical results to those observed with OA (Supplementary Fig. 5d, e). Thus, inhibition of cellular PP2A activity augments EGF-induced IEG expression mainly through the enhancement of NELF-A phosphorylation. Consistent with these results, H1299 and A375 cancer cells exhibited very low IEG expression at steady-state (where NELF-A remains dephosphorylated despite high basal ERK activity) (Fig. 7b). However, when these cells were treated with one of the PP2A inhibitors (OA, rubratoxin A, or cytostatin) to increase NELF-A phosphorylation, marked induction of the IEGs was observed (Fig. 7b and Supplementary Fig. 5f–h). Thus, PP2A counteracts NELF-A phosphorylation induced by ERK-activating oncogenes, and represses IEG expression in cancer cells. Collectively, these findings indicate that oncogene-induced, constitutive ERK activity alone is insufficient to provoke aberrant NELF-A phosphorylation and IEG expression in cancer cells. Rather, increased ERK activity must be accompanied by decreased PP2A activity. In other words, NELF-A is a unique ERK substrate in that its phosphorylation status is strictly controlled by PP2A in cancer.

### PP2A inhibitor proteins promote tumor progression by upregulating IEGs and growth-promoting genes in human cancers

Recent clinical sequencing studies identified gain-of-function mutations in various ERK-activating oncogenes as well as aberrant overexpression of PP2A inhibitor proteins, such as SET and CIP2A, in human cancers[6,32].

However, their precise functional relationship remains elusive. Our results imply that simultaneous ERK activation and PP2A inhibition would synergize to increase NELF-A phosphorylation and IEG expression, which in turn might accelerate tumor progression. To test this hypothesis, we first determined whether overexpression of SET or/and CIP2A co-occurred with mutations of ERK-activating oncogenes in human cancers. Analyses of large-scale cancer genomics datasets showed significantly higher expression of SET and CIP2A in various cancers than in their corresponding normal tissues (Fig. 8a, b). Furthermore, SET/CIP2A overexpression was frequently observed in cancers with mutations in various oncogenes (e.g., *EGFR*, *KRAS*, *NRAS*, and *BRAF*) (Fig. 8a, b and Supplementary Fig. 6a–d). Thus, the aberrant PP2A inhibition by SET/CIP2A and ERK hyperactivation by oncogenes coexist in many clinical cancer cases. We then investigated whether overexpression of the PP2A inhibitor proteins is relevant to the upregulation of IEGs and growth-promoting genes in cancers with an ERK-activating oncogene. Analysis of a large cohort of KRAS-mutated pancreatic cancer cases revealed that the mRNA levels of various IEGs and growth-promoting genes (i.e., the G1/S cyclins D1 and E2, and the cell proliferation marker, PCNA) were substantially increased in the SET-high (top 10%) cancers compared with the SET-low (bottom 10%) cancers (Fig. 8c, left). Similar results were also observed when the pancreatic cancers were divided into two groups based on their CIP2A expression levels (Fig. 8c, right). Furthermore, scatter plot analyses showed significant positive correlations of SET/CIP2A expression with the expression levels of the IEGs (*TRIB1*, *SPRY4*, *CTGF*) and with those of growth-promoting genes (*cyclin D1*, *E2*, and *PCNA*) in KRAS-mutated pancreatic cancers (Fig. 8d, e). Consistent with these results, SET overexpression is associated with poor clinical prognosis in patients with pancreatic cancers (Fig. 8f). We confirmed that enhanced expression of SET or CIP2A is sufficient to promote strong NELF-A phosphorylation (Fig. 8g) and IEG expression (Fig. 8h) in H1299 cancer cells harboring NRAS[Q61K]. Thus, aberrant overexpression of SET/CIP2A, in concert with ERK-activating oncogenes, strongly upregulates IEGs and their downstream growth-promoting genes in human cancers. This occurs mainly through the induction of NELF-A phosphorylation and the resulting Pol-II pause release, thereby accelerating tumor progression.

### Discussion

In this study, we identified NELF-A as an ERK substrate, and demonstrated that ERK-mediated NELF-A phosphorylation is crucial for growth factor-induced Pol-II pause release and the resulting productive transcription elongation of IEGs, thereby promoting cell growth. While the importance of ERK in the transcription initiation step (i.e., phosphorylation-dependent activation of TFs) is well documented[1], less is known about the role of ERK in the regulation of the elongation process. We found that ERK-dependent phosphorylation of NELF-A facilitates the dissociation of NELF from the stalled pol-II at the promoter-proximal regions of IEGs, thereby releasing pol-II into productive elongation. Although P-TEFb is known to phosphorylate NELF-A mainly at T[157], T[277], and S[363] at least in vitro[16,17], the precise mechanisms that regulate in vivo NELF-A phosphorylation have

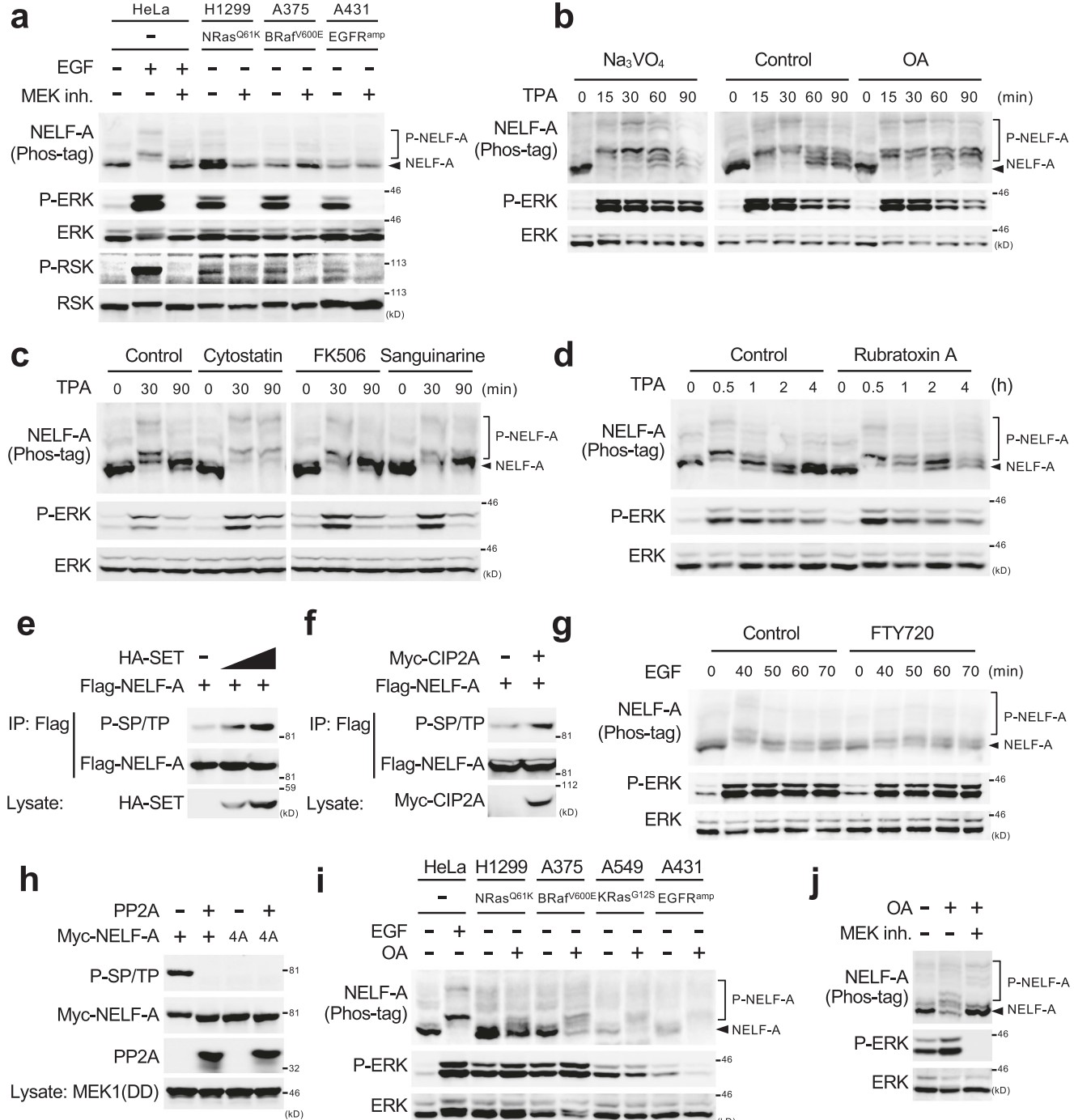

**Fig. 6 | PP2A counteracts oncogene-induced NELF-A phosphorylation in cancer cells. a** Cells were treated with or without the MEK inhibitor, trametinib. The phosphorylation status of endogenous NELF-A was analyzed using Phos-tag SDS-PAGE followed by immunoblotting with an anti-NELF-A antibody. HeLa cells + EGF served as a positive control. **b–d** HeLa cells were pretreated with an inhibitor of dual-specificity and tyrosine phosphatases (Na$_3$VO$_4$, 500 μM), a PP2A inhibitor (OA, 400 nM), or control solvent (DMSO) (**b**), with a selective PP2A inhibitor (Cytostatin, 20 μM), a PP2B inhibitor (FK506, 0.2 μM), or a PP2C inhibitor (Sanguinarine, 0.5 μM) (**c**), or with a specific PP2A inhibitor (Rubratoxin A, 20 μM) (**d**) for 60 min, and stimulated with TPA (300 nM) for the indicated times. The phosphorylation status of endogenous NELF-A was monitored as in **a. e, f** HEK293 cells were transfected with Flag-NELF-A, together with HA-SET (**e**) or Myc-CIP2A (**f**). Phosphorylation of immunoprecipitated Flag-NELF-A was probed with an anti-phospho-SP/TP antibody (top). Total protein levels are shown in the lower rows. **g** HeLa cells were pretreated with or without a PP2A-activating drug (FTY720, 2.5 μM) for 60 min, and stimulated with EGF for the indicated times. The phosphorylation status of NELF-A was monitored as in **a. h** Phosphorylated Myc-NELF-A was immunopurified from HEK293 cell co-expressing MEK1(DD), and incubated with or without purified recombinant PP2A in vitro. The phosphorylation status of Myc-NELF-A was assessed by immunoblotting with an anti-phospho-SP/TP antibody. **i, j** In (**i**), control HeLa cells and cell lines harboring the indicated oncogenes were treated with or without OA (400 nM, for 60 min). EGF-stimulated HeLa cells served as a positive control for NELF-A phosphorylation. In (**j**), H1299 cells were treated with OA in the presence or absence of the MEK inhibitor (trametinib). **i, j** Phosphorylation of endogenous NELF-A was assessed as in **a.** Source data are provided as a Source Data file.

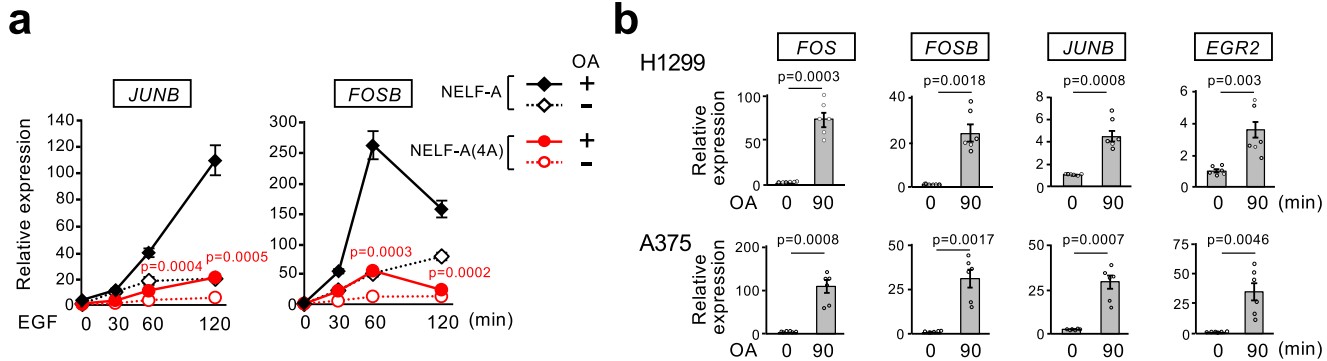

**Fig. 7 | PP2A-mediated dephosphorylation of NELF-A suppresses IEG expression. a, b** In (**a**), HEK293-WT (black lines) and -4A (red lines) cells were stimulated with EGF (20 ng/ml) for the indicated times in the presence or absence of OA (400 nM). In (**b**), H1299 and A375 cells were treated with OA for 90 min. **a, b** Levels of the indicated IEG mRNAs were quantified using qRT-PCR. Data were normalized to *GAPDH* expression. Error bars, SEM (*n* = 6). *P*-values were assessed using a two-tailed Student's *t*-test. Source data are provided as a Source Data file.

remained obscure. Furthermore, recent studies have reported P-TEFb-independent dissociation of NELF from chromatin[18–20]. Therefore, the functional significance of P-TEFb in stimulus-induced NELF dissociation from chromatin is still unclear. In the present study, we showed that upon growth factor stimulation, ERK phosphorylated NELF-A at $S^{363}$, $S^{393}$, $T^{396}$, and $T^{399}$. These sites are mostly different from those phosphorylated by P-TEFb, and we also observed that CDK9 depletion did not suppress the phosphorylation. Therefore, our data indicate that growth factor-induced NELF-A phosphorylation is mediated by ERK but not by P-TEFb in vivo, and suggest that NELF-A is phosphorylated in a context-dependent manner.

On the basis of the present results, we propose the following mechanism for growth factor-induced, ERK-mediated IEG expression (Fig. 9, left). Under steady-state conditions (where ERK activity is low), shortly after the transcription initiation, NELF (including the dephosphorylated form of NELF-A), in concert with DSIF, binds to and stalls Pol-II at the promoter-proximal regions of IEGs. The resulting PPP of Pol-II suppresses productive mRNA synthesis of IEGs. However, upon growth factor stimulation, activated ERK rapidly phosphorylates NELF-A, mainly at $S^{363}$, $S^{393}$, $T^{396}$, and $T^{399}$. At the same time, P-TEFb is recruited to promoter-proximal regions, where it phosphorylates the Pol-II CTD at $Ser^2$ and DSIF[8]. These ERK and P-TEFb-mediated multiple phosphorylation events cooperatively induce NELF dissociation from Pol-II and elicit the transition of Pol-II from pausing into productive elongation, thereby inducing IEG expression. ERK also phosphorylates and activates TFs responsible for IEG expression (e.g., ELK1[2], which further increases the rate of the aforementioned transcription cycle. Through these coordinated actions, ERK rapidly and strongly upregulates IEG expression in response to mitogenic stimuli and drives cell cycle progression. Our data reveal a previously unexplored role of ERK in the regulation of IEG expression and the resulting cell growth. Since the ERK phosphorylation sites of NELF-A are evolutionarily conserved among vertebrates, this mechanism of regulating Pol-II elongation might be similarly conserved across species and control growth factor-induced expression of genes with high pausing indices.

Another important finding is that PP2A efficiently dephosphorylates NELF-A and thereby rigorously controls the expression levels of IEGs under physiological and pathological conditions. Following TPA stimulation, NELF-A phosphorylation gradually declined and returned to a basal level by about 2 h, even though high ERK activity persisted for more than 4 h. This NELF-A dephosphorylation was repressed by the PP2A inhibitors, but accelerated by a PP2A activator. Thus, the phosphorylation state of NELF-A is principally determined by the balance between ERK and PP2A activities, and its phosphorylation requires a relatively high level of ERK activity to overcome PP2A-mediated dephosphorylation. Furthermore, we

showed that suppression of cellular PP2A activity dramatically augmented growth factor-induced IEG expression mainly through enhancing NELF-A phosphorylation. Thus, PP2A-mediated control of the magnitude and duration of NELF-A phosphorylation is vital for the proper regulation of IEG expression. In other words, robust PP2A-mediated NELF-A dephosphorylation serves to buffer ERK activity in order to prevent cells from inadvertent or excessive IEG induction and cell growth.

Consistent with this notion, we found here that the constitutive (but relatively moderate) ERK activation induced by oncogenes is insufficient to induce NELF-A phosphorylation. Importantly, however, chemical or genetic suppression of cellular PP2A activity readily induced NELF-A phosphorylation, indicating that PP2A activity is critical in determining the phosphorylation state of NELF-A in cancer cells. Thus, NELF-A is unique among ERK substrates in that its phosphorylation is not simply induced by oncogene-induced ERK activation; rather, it also requires concurrent PP2A suppression in cancer. This PP2A-dependent control of NELF-A phosphorylation provides vital insight into the roles of the PP2A inhibitor proteins in tumorigenesis (Fig. 9, right). In cancer cells with high ERK activity but intact PP2A, ERK can phosphorylate its substrate TFs, but not NELF-A, as PP2A blocks NELF-A phosphorylation. Therefore, oncogene-induced ERK activity often fails to release Pol-II from PPP into productive elongation of IEGs. However, in cancer cells with high ERK activity and high PP2A inhibitor protein (SET/CIP2A) expression, the decreased PP2A activity leads to increased NELF-A phosphorylation and consequent productive mRNA synthesis of IEGs, thereby promoting cancer development and progression. Indeed, in human pancreatic cancers with KRAS mutations, SET/CIP2A overexpression correlates with the increased expression of IEGs, resulting in poor clinical outcomes. Thus, SET/CIP2A-mediated PP2A suppression in cancer is particularly important for the induction of NELF-A phosphorylation. It may even act as an on-off switch in determining the phosphorylation of NELF-A (and probably other ERK substrates that are efficiently dephosphorylated by PP2A) and the subsequent expression of IEGs. Our results delineate the unique role of the PP2A inhibitor proteins in tumorigenesis, and potentially explain why SET and CIP2A are frequently overexpressed concomitantly with mutations in ERK-activating oncogenes in human cancers. Recently, several chemical activators of PP2A have been developed and tested for cancer treatment, although they are still at the preclinical stage of development[6]. Considering that PP2A activity is a key determinant of NELF-A phosphorylation in cancer, the pharmacological reactivation of PP2A would be highly effective in suppressing the expression of IEGs and growth-promoting genes, and thus it may provide significant benefits in cancer treatment.

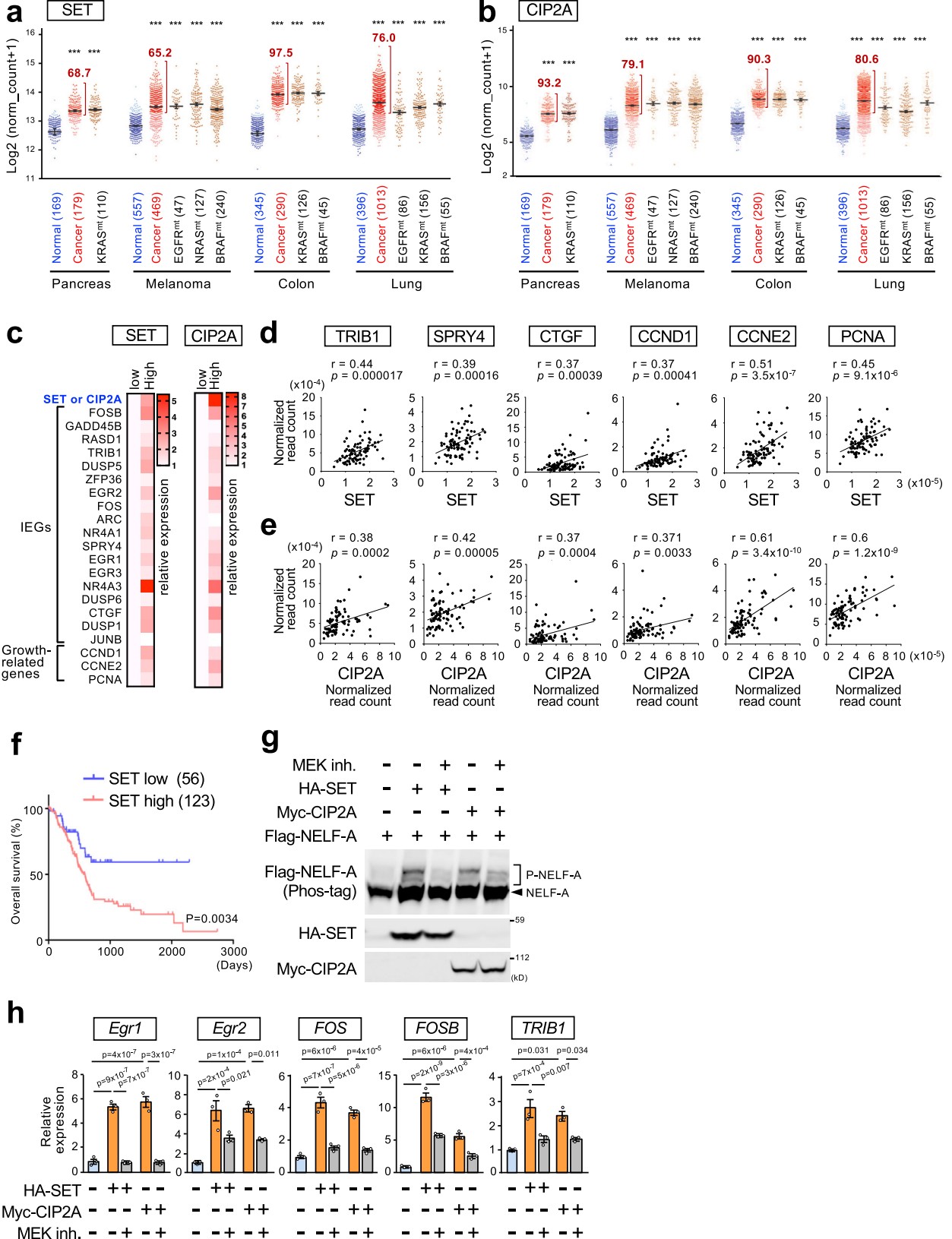

## Methods

### Media and buffers

Lysis buffer A contained 20 mM Tris-HCl (pH 7.5), 1% Triton X-100, 0.5% deoxycholate, 10% glycerol, 150 mM NaCl, 2 mM EDTA (pH 8.0), 50 mM β-glycerophosphate, 10 mM NaF, 2 mM $Na_3VO_4$, 1 mM DTT, 1 mM phenylmethylsulphonyl fluoride (PMSF), 10 μg/ml leupeptin, and 10 μg/ml aprotinin. Lysis buffer B contained 20 mM Tris-HCl (pH 7.5), 0.5% Triton X-100, 10% glycerol, 137 mM NaCl, 2 mM EDTA (pH 8.0), 50 mM β-glycerophosphate, 10 mM NaF, 1 mM $Na_3VO_4$, 1 mM DTT, 1 mM PMSF, 10 μg/ml leupeptin, and 10 μg/ml aprotinin. Lysis buffer C contained 20 mM Tris-HCl (pH 7.5), 1% Triton X-100, 0.1% deoxycholate, 10% glycerol, 150 mM NaCl, 50 mM NaF, 1 mM DTT, 1 mM

**Fig. 8 | PP2A inhibitor proteins promote tumor progression by upregulating growth-promoting genes in human cancers. a, b** Analyses of SET (**a**) and CIP2A (**b**) mRNA expression in normal, cancer tissues, and cancer subsets with mutations in the indicated oncogenes based on the TCGA-TARGET-GTEx databases. Each horizontal bar represents the mean. Error bars, SEM. Numbers in parentheses indicate sample sizes. Numbers on the graph represent the percentage of patients with SET- or CIP2A-high cancers. Cancers in which SET or CIP2A expression is higher than the top 5% level of the corresponding normal tissues are defined as SET- or CIP2A-high. *P*-values were assessed using a one-way ANOVA Tukey test. ***$p < 1.4 \times 10^{-10}$. **c** Relative expression levels (fold change) of IEGs and growth-promoting genes in KRAS-mutated pancreatic cancers with high (top 10%; $n = 9$) and low (bottom 10%; $n = 9$) expression of SET or CIP2A. A total of 87 samples with RNAseq data were obtained from the ICGC dataset. **d, e** Scatter plots showing the

Pearson's correlation of SET (**d**) or CIP2A (**e**) expression with the levels of IEGs (*TRIB1*, *SPRY4*, and *CTGF*) and growth-promoting genes (*CCND1*, *CCNE2*, and *PCNA*) in KRAS-mutated pancreatic cancers ($n = 87$). Pearson's correlation coefficients (*r*) and two-tailed *P*-values were determined. **f** Kaplan–Meier curves showing the overall survival of TCGA pancreatic cancer patients with high ($n = 123$) or low ($n = 56$) SET expression. **g, h** H1299 cells were transfected with Flag-NELF-A, together with HA-SET or Myc-CIP2A. Where indicated, the cells were treated with the MEK inhibitor, trametinib. In (**g**), phosphorylation of Flag-NELF-A was analyzed using Phos-tag SDS-PAGE followed by immunoblotting. In (**h**), expression levels of the indicated IEG mRNAs were analyzed using qRT-PCR. Error bars, SEM ($n = 3$). *P*-values were assessed using a one-way ANOVA Tukey test. Source data are provided as a Source Data file.

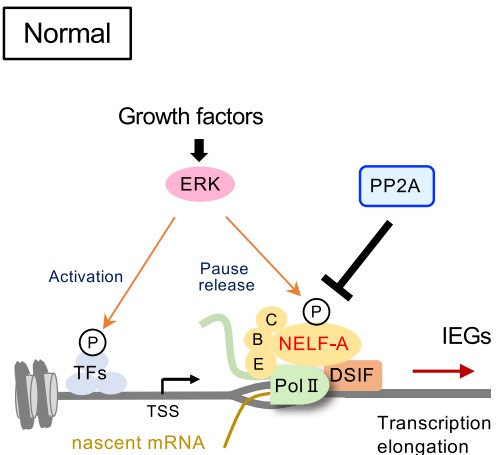

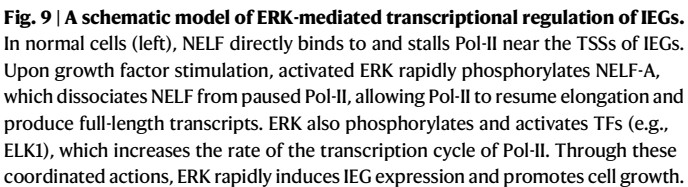

**Fig. 9 | A schematic model of ERK-mediated transcriptional regulation of IEGs.** In normal cells (left), NELF directly binds to and stalls Pol-II near the TSSs of IEGs. Upon growth factor stimulation, activated ERK rapidly phosphorylates NELF-A, which dissociates NELF from paused Pol-II, allowing Pol-II to resume elongation and produce full-length transcripts. ERK also phosphorylates and activates TFs (e.g., ELK1), which increases the rate of the transcription cycle of Pol-II. Through these coordinated actions, ERK rapidly induces IEG expression and promotes cell growth.

In contrast, PP2A dephosphorylates NELF-A, thereby preventing cells from excessive IEG production. Therefore, in cancer cells (right) with intact PP2A, the relatively moderate ERK activity induced by oncogenes fails to induce NELF-A phosphorylation. However, when PP2A activity is repressed by overexpression of the PP2A inhibitors (SET and CIP2A), the decreased PP2A activity combined with oncogene-induced constitutive ERK activity conspire to induce NELF-A phosphorylation and IEG expression, thereby promoting cancer development and progression.

PMSF, 10 µg/ml leupeptin, 10 µg/ml aprotinin, and 100 nM microcystin LR. Kinase buffer contained 25 mM Tris-HCl (pH 7.5), 25 mM MgCl$_2$, 7 mM β-glycerophosphate, 0.5 mM Na$_3$VO$_4$, and 2 mM DTT. Phosphatase buffer contained 25 mM Tris-HCl (pH 7.5), 1 mM EDTA, and 1 mM DTT.

## Plasmids

The human NELF-A cDNA (accession No., BC002764.2) was obtained from Open Biosystems. Myc-NELF-A, Flag-NELF-A, His-NELF-A, Flag-NELF-B, Flag-NELF-C, Flag-NELF-E, HA-SET, Myc-CIP2A, HA-MEK1, Myc-MEK1, HA-ERK2, HRas(G12V), and their derivative mutants were subcloned into pcDNA3 (Invitrogen), pQCXIP, pQCXIH (Clontech), or pWZL-Blast (Addgene). Various point mutants of NELF-A were constructed using PCR-based mutagenesis. A constitutively active MEK1(DD) and a kinase-inactive ERK2(K52N) were described previously[33]. pSUPER.retro.puro vector (Oligoengine) was used to generate the NELF-A-shRNA construct (see Supplementary Table 1). pGEX6P-1 (GE Healthcare) was used for bacterial expression of GST-NELF-A. A yeast expression plasmid pLexA-WW was constructed by inserting the WW domain of human Pin1(amino acids 6-39) in frame with LexA into pLexA. p426ADH-ERK2(PD) was described previously[4].

## Cell culture and treatment

A375 (CRL-1619) and H1299 (CRL-5803) were obtained from ATCC. HEK293 (RCB1637), COS7 (RCB0539), HeLa (RCB0007), A549 (RCB3677), A431 (RCB0202), and GP2-293 (RCB2354) were obtained

from RIKEN cell bank. HaCaT cells were kindly provided by Prof. Dr. N. Fusenig (German Cancer Research Center, Germany). Cells were maintained in Dulbecco's modified Eagle medium supplemented with 10% fetal bovine serum, L-glutamate, penicillin, and streptomycin. Where indicated, the cells were stimulated with EGF (20 ng/ml) or TPA (300 nM) for 15 min or the indicated times. Before stimulation, cells were pre-treated with MEK or ERK inhibitors [10 µM trametinib (LC Laboratories), 10 µM AZD6244 (LC Laboratories), or 10 µM FR180204 (Cayman)] for 45 min, a Cdk9 inhibitor [100 µM DRB (Cayman)] for 3 h, or with phosphatase inhibitors [400 nM okadaic acid, 0.2 µM FK506 (Wako), 20 µM cytostatin, 20 µM rubratoxin A (Institute of Microbiol Chemistry), or 0.5 µM sanguinarine (Cayman)] for 60 min as indicated, unless otherwise noted.

## Yeast three-hybrid analysis

Yeast three-hybrid analysis was performed using a yeast TM414 (L40 *ura3*) strain[4]. TM414 cells were first transformed with the pLexA-WW bait together with a p426ADH-ERK2(PD) expression plasmid. The cells were then further transformed with a mixture of human cDNA libraries (derived from liver, placenta, and brain) constructed in the pACT2 vector. Transformants were grown on selective medium (synthetic complete medium without His, Leu, Ura, and Trp) containing 3-aminotriazole (1 mM). Yeast colonies exhibiting histidine-independent growth were isolated and further analyzed. For the yeast spot assay, yeast cells were grown on a selection medium containing 3-aminotriazole (1 mM).

## Transient transfection and generation of stable cell lines

For transient transfection, pre-seeded cells were transfected with the appropriate expression plasmids using the X-tremeGENE 9 DNA reagent (Sigma-Aldrich) according to the manufacturer's protocol. For siRNA transfection experiments, cells were transfected with an siRNA targeting CDK9 (see Supplementary Table 1), INTS6 (siRNA ID: SASI_Hs01_00121637, SASI_Hs01_00229908, Sigma), or MISSION siRNA Universal Negative Control (Sigma-Aldrich) using Lipofectamine RNAiMAX (Thermo Fisher). To generate HEK293/HaCaT-WT and −4A cells, parental HEK293 or HaCaT cells were first transfected with a pQCXIH-Myc-NELF-A(WT or 4A) expression vector, in which the coding sequence for the insert (shRNA-resistant Myc-NELF-A) is co-transcribed with the hygromycin resistance gene as a bicistronic message via an internal ribosome entry site. Pools of cells stably expressing Myc-NELF-A(WT or −4A) were selected with hygromycin. Endogenous NELF-A was then stably knocked down by retroviral expression of shRNA specific for NELF-A. Infected cells were selected with puromycin, pooled, and assayed for Myc-NELF-A protein expression by immunoblotting. HaCaT-WT and −4A cell lines stably expressing HRas(G12V) were generated by transfection of a pWZL-Blast-HRas(G12V) expression vector and blasticidin selection.

## Immunoprecipitation and co-immunoprecipitation assays

For immunoprecipitation experiments, cell lysates were prepared with lysis buffer A and precleared with protein G-Sepharose beads (GE Healthcare) at 4 °C for 1 h. Precleared lysates were incubated with an appropriate antibody [anti-Flag M2 (Sigma, F1804; 3 μg/sample) or anti-Myc 9E10 (Santa Cruz, sc-40; 1 μg/sample) antibody] for 3 h with gentle rotation. Protein G-Sepharose beads were then added and incubated for 1 h. Immunoprecipitates were collected by centrifugation and washed three times with lysis buffer A containing 0.1% SDS and subjected to SDS-PAGE. For co-immunoprecipitation assays, cells were harvested with lysis buffer B and sonicated using Qsonica Q700 (QSonica) and centrifuged. The supernatants were incubated with an appropriate antibody [anti-Myc 9E10 antibody (Santa Cruz, sc-40; 1 μg/sample) or mouse control IgG$_1$ G3A1 (CST, 5415; 1 μg/sample)] overnight at 4 °C. Protein G-Dynabeads (Thermo Fisher) were then added and incubated for 1 h. Immunoprecipitates were washed three times with lysis buffer B. Proteins were separated by SDS-PAGE and immunoblotted with the indicated antibodies.

## Purification of recombinant NELF-A

Recombinant GST-NELF-A was expressed in the *E. coli* strain DH5α using pGEX6P-1 vector, and purified using glutathione-Sepharose beads (GE Healthcare) as previously described[33]. Briefly, DH5α was transformed with a pGEX-6P-based plasmid encoding GST-NELF-A. Exponentially growing DH5α cells were incubated with 0.5 mM IPTG at 25 °C for 14 h before harvesting. Cells were suspended in cold PBS, lysed by sonication, and then clarified by centrifugation (30,000 × *g* for 15 min at 4 °C). The clear supernatant was filtered through a 0.45 μm filter, and GST-NELF-A proteins were purified using glutathione-Sepharose beads and used for an in vitro kinase assay.

## In vitro kinase and phosphatase assays

For an in vitro kinase assay, HEK293 cells were cotransfected with HA-ERK2 (WT or K52N) and Myc-MEK1(DD) and lysed in lysis buffer A. Cell lysates were incubated with anti-HA antibody (Roche, 11867423001; 1 μg/sample) at 4 °C for 3 h. Immune-complexes were recovered with the aid of protein G-Sepharose beads, and were washed three times with lysis buffer A without DOC and then twice with kinase buffer. Immunoprecipitates were resuspended in 40 μl of kinase buffer containing 5 μg of GST-NELF-A. The kinase reaction was initiated by the addition of 0.1 mM ATP. Following 30-min incubation at 30 °C, reactions were

terminated by the addition of SDS loading buffer. For the in vitro phosphatase assay, phosphorylated Myc-NELF-A was immunoprecipitated using anti-Myc 9E10 antibody (Santa Cruz, sc-40; 1 μg/sample) from HEK293 cells co-expressing HA-MEK1(DD) and the immunoprecipitate was washed and resuspended in 40 μl of phosphatase buffer. Following the addition of 1 μg recombinant PP2A proteins (P16-20BH, SignalChem), reaction mixtures were incubated at 37 °C for 2 h.

## Immunoblotting analyses

Immunoblotting analyses were carried out as described previously[4]. Briefly, appropriate amounts of proteins were resolved by SDS–PAGE and transferred onto nitrocellulose membranes. After blocking with 4% skim milk, membranes were probed with appropriate antibodies and visualized using enhanced chemiluminescence detection. Digitized images were captured by ImageQuant LAS4000 (GE Healthcare). The following primary antibodies were used: monoclonal anti-NELF-A G-11 (Santa Cruz, sc-365004), anti-NELF-D C-10 (sc-393972), anti-NELF-E F-9 (sc-377052), anti-ERK1/2 C-9 (sc-514302), anti-Myc 9E10 (sc-40), anti-GST B-14 (sc-138), anti-HA F-7 (sc-7392), anti-Pol II 8WG16 (sc-56767), anti-Pol II CTD4H8 (sc-47701), anti-CDK9 D-7 (sc-13130), anti-Elk1 E-5 (sc-365876), anti-CyclinD1 A-12 (sc-8396); anti-Phospho-Rpb1 CTD-Ser5 D9N5I (CST, 13523), anti-RSK1/RSK2/RSK3 32D7 (9355), anti-Phospho-Rpb1 CTD-Ser2 E1Z3G (13499), anti-c-Fos 9F6 (2250); anti-HA 3F10 (Roche, 11867423001); anti-Flag M2 (Sigma, F1804); anti β-Actin (Wako, 010-27841); anti-His-tag (Medical & Biological Laboratories, D291-3); anti-PP2Ac (BD, 610555), anti-DICE1/INTS6 (Santa Cruz Biotechnology, sc-376524), anti-MPM2 (anti-phospho-SP or TP) (Millipore, 05-368); polyclonal anti-ERK1 K-23 (Santa Cruz, sc-94); anti-NELF-A (Protein tech, 10456-1-AP), anti-NELF-B (16418-1-AP); anti-phospho-ERK1/2 (CST, 9101), anti-Phospho-p90RSK T573 (9346); anti-GADD45B (Cloud-Clone, PAL535Hu01), anti-RAS (G12V mutant) (GeneTex, GTX132694). All antibodies were used at a dilution of 1:1000, except anti-Pol II (1:2000), anti-Phospho-Rpb1 (1:2000), anti-NELF-A (1:3000), anti-NELF-B (1:2000), anti-NELF-D (1:2000), anti-NELF-E (1:2000), and anti-CDK9 (1:2000). The following secondary antibodies were used: anti-mouse IgG-horse radish peroxidase (HRP) antibody (1:5000, NA931, Cytiva), and anti-rabbit IgG-HRP antibody (1:2500, NA934, Cytiva). In Fig. 3a and Supplementary Fig. 3a, the band intensities were quantified using ImageJ software (ver. 1.53a).

## Phos-tag SDS-PAGE and Pro-Q diamond gel staining

For Phos-tag SDS-PAGE analysis, cell lysates were prepared with lysis buffer C and separated with 8% SDS-PAGE gels containing 50 μM Phos-tag acrylamide (Wako) and 0.1 mM MnCl$_2$. After electrophoresis, gels were washed three times with transfer buffer (25 mM Tris, 192 mM glycine, 20% methanol) containing 10 mM EDTA, and then washed once with transfer buffer without EDTA according to the manufacturer's protocol. The separated proteins were transferred to a nitrocellulose membrane and probed with the appropriate antibodies. For pro-Q Diamond gel staining, immunoprecipitated NELF-A proteins were first separated by conventional SDS-PAGE, and the gel was stained with ProQ Diamond Phosphoprotein Gel Stain (Thermo Fisher) according to the manufacturer's protocol. Total NELF proteins were then detected using SilverQuest (Invitrogen) or SYPRO Ruby (Thermo Fisher) staining methods as indicated.

## Mass spectrometry

HEK293 cells stably expressing Flag-NELF-A were stimulated with TPA (300 nM, for 45 min) and then harvested in lysis buffer A. After immunoprecipitation with anti-Flag M2 antibody (Sigma, F1804; 3 μg/sample), the immune complexes were washed three times with lysis buffer A containing 0.1% SDS and twice with MS wash buffer (20 mM Tris HCl pH 7.5, 150 mM NaCl). Phosphorylated Flag-NELF-A was eluted with Flag peptide (Sigma-Aldrich), digested with trypsin,

desalted using ZipTip C18 (Millipore), and centrifuged in a vacuum concentrator. Shotgun proteomic analyses were performed using a linear ion trap-orbitrap mass spectrometer (LTQ-Orbitrap Velos, Thermo Fisher) coupled with a nanoflow LC system (Dina-2A, KYA Technologies) ($n = 1$). Peptides were injected into $75\,\mu m$ reversed-phase C18 column at a flow rate of $10\,\mu L/min$ and eluted with a linear gradient of solvent A (2% acetonitrile and 0.1% formic acid in $H_2O$) to solvent B (40% acetonitrile and 0.1% formic acid in $H_2O$) at $300\,nL/min$. The peptides sprayed from nano electrospray ion source (KYA Technologies, Tokyo, Japan) were analyzed by the collision-induced dissociation (CID) method. The analyses were operated in data-dependent mode, switching automatically MS and MS/MS acquisition. All full-scan MS spectra in the range from $m/z$ 380 to 2000 were acquired in the orbitrap with a resolution of 100,000 at $m/z$ 400 after ion count accumulation to the target value of 1,000,000. The 20 most intense ions that satisfied an ion selection threshold above 2000 were fragmented in the linear ion trap with a normalized collision energy of 35% for an activation time of 10 ms. The orbitrap analyzer was operated with the lock mass option to carry out shotgun detection with high accuracy. Phosphorylation sites were determined by database searches with Mascot (Matrix Science, ver. 2.4) and Proteome Discoverer (Thermo Fisher Scientific, ver. 1.3).

### Immunofluorescence staining

Cells growing on glass coverslips were stimulated with or without EGF (20 ng/ml), fixed with 3% paraformaldehyde in PBS for 10 min, and permeabilized with 0.1% Triton X-100 for 5 min. After washing with PBS, the cells were incubated in the blocking solution BlockAce (Snow Brand Milk Products) for 1 h. Cells were then incubated with anti-Myc antibody 9E10 (Santa Cruz, sc-40; 1 μg/ml) overnight at 4 °C in PBS containing 2% BSA. After washing four times, the cells were incubated with Alexa-488 goat anti-mouse antibody (Molecular Probes, A-11029; 1:2000) for 30 min. The stained cells on coverslips were washed three times with PBS and mounted in FluorSave Reagent (Calbiochem). A Nikon Eclipse Ti-fluorescent microscope equipped with a Rolera EM-C2™ EMCCD camera (QImaging) and the Universal Metamorph software (Molecular Devices) were used to capture fluorescence microscopic images.

### RNA sequencing and data analysis

HEK293-WT or −4A cells were stimulated with or without EGF (20 ng/ml for 60 min). Total RNA was isolated using the RNeasy Mini Kit (QIAGEN). RNA libraries were prepared using 100 ng of total RNA with an Ion AmpliSeq Transcriptome Human Gene Expression kit (Thermo Fisher) according to the manufacturer's instructions. The libraries were sequenced on an Ion Proton system using an Ion PI Hi-Q Sequencing 200 kit and Ion PI Chip v3 (Thermo Fisher), and the sequencing reads were aligned to hg19_AmpliSeq_Transcriptome_ERCC_v1 using Torrent Mapping Alignment Program (TMAP). The data were then analyzed using AmpliSeqRNA plug-in (Thermo Fisher, ver. 5.2.0.3), a Torrent Suite Software (Thermo Fisher, ver. 5.2.2), which provides QC metrics and normalized read counts per gene. The volcano plots and heatmap were generated using R packages (gplots or ggplot2) based on the normalized RPM values. The RNA-seq data in this study were deposited in the Gene Expression Omnibus (GEO) database under accession number GSE167233.

### Quantitative reverse transcription-PCR analyses (qRT-PCR)

Total RNA was extracted using TRIzol reagent (Thermo Fisher) according to the manufacturer's protocol. One μg of RNA was reverse-transcribed using the PrimeScript-RT Master Mix (TaKaRa). Real-time qRT-PCR analysis was performed using the Thermal Cycler Dice real time system (TaKaRa) or CFX Connect system (BIO-RAD), together with Thunderbird SYBR qPCR mix reagent (Toyobo). The relative gene expression levels were calculated by normalizing to that of *GAPDH*. The sequences of primers used for qRT-PCR are listed in Supplementary Table 1.

### ChIP-qPCR assay

The ChIP assay was performed using the SimpleChIP Enzymatic Chromatin IP kit (CST) according to the manufacturer's protocol. Immunoprecipitates were recovered with Protein-G Dynabeads (Invitrogen) together with specific antibodies [anti-Myc 9B11 (CST, 2276; 0.4 μg/sample), anti-Pol II CTD 4H8 (Santa Cruz, sc-47701; 2 μg/sample), or mouse control $IgG_1$ G3A1 (CST, 5415; 1 μg/sample)] at 4 °C for 12 h with gentle rotation. The purified DNA was subjected to quantitative PCR analysis using Thunderbird SYBR qPCR mix (Toyobo). All primers used were listed in Supplementary Table 1.

### Cell proliferation assay and mouse xenograft

For the cell proliferation assay, $3 \times 10^3$ HaCaT cells/well were seeded in triplicate in DMEM with 10% FBS, and cell growth was monitored each day using the Cell Counting Kit-8 (Dojindo). For the mouse xenograft experiments, five-week-old female BALB/c-nude mice were purchased from Oriental Yeast (Tokyo, Japan). $5 \times 10^6$ HaCaT-WT, HaCaT-4A, Ras-HaCaT-WT, or Ras-HaCaT-4A cells suspended in 100 μl Matrigel (Corning) were injected subcutaneously into the left or right side flank of each mouse. Tumor volume ($mm^3$) was measured with calipers and was calculated as $(L \times W^2)/2$, where L is length and W is width. This study was conducted in accordance with the Regulations for Animal Care and Use of The University of Tokyo and the Guidelines for Proper Conduct of Animal Experiments by the Science Council of Japan, and was approved by the Animal Experiment Committee at the Institute of Medical Science, The University of Tokyo (approval number: A18-47). The housing conditions for the mice are as follows: temperature $22 \pm 2$ °C, humidity $55 \pm 5\%$, and light/dark cycle 12 h/12 h (8 a.m.–20 p.m. light).

### BrdU incorporation assay

BrdU assay was performed using 5-bromo-2′-deoxy-uridine Labeling and Detection Kit I (Sigma-Aldrich, catalog #11296736001) according to the manufacturer's instructions. Briefly, $1 \times 10^5$ cells were serum-starved for 72 h on a coverslip in a 35-mm dish, and then stimulated with 100 ng/ml EGF for 18 h. After incubation with 10 μM BrdU for 1 h, cells were fixed and incubated with an anti-BrdU antibody (1:10 dilution of Anti-BrdU stock solution supplied by the manufacturer) and a secondary anti-mouse-Ig-fluorescein antibody (1:10 dilution of Anti-mouse-Ig-fluorescein stock solution supplied by the manufacturer). The nuclei were labeled with DAPI. The percentage of BrdU-positive cells was analyzed using Operetta CLS (PerkinElmer) equipped with Harmony software (ver. 4.5).

### In silico analyses of gene expression data from human tumors

To compare the gene expression levels between normal and tumor tissues, the gene expression and clinical data of pancreas, skin, lung, and colon cancer patients were retrieved from a combined cohort of TCGA, TARGET, and GTEx samples (TCGA-TARGET-GTEx), and from the International Cancer Genome Consortium (ICGC) using the UCSC Xena Browser (https://xenabrowser.net/), and were analyzed using GraphPad Prism9. Genetic alterations and gene expression in the TCGA clinical samples were visualized as an Oncoprint plot using cBioportal (https://www.cbioportal.org/).

### Statistics and reproducibility

The statistical significance of the difference between mean values was tested by a two-tailed Student's *t*-test or one-way ANOVA Tukey test using GraphPad Prism. Data are presented as means ± SEM.

Pearson's correlation coefficients were calculated to assess the correlation between the expression levels of PP2A inhibitors (SET and CIP2A) and the growth-related genes. All the experiments shown in Figs. 1c–i, 2c, e, 3c, and 6b–j, and Supplementary Figs. 1c, d, 2b, 3a–c, 4a, b, 5a, and b were repeated independently at least three times with similar results.

## Reporting summary
Further information on research design is available in the Nature Portfolio Reporting Summary linked to this article.

## Data availability
The data that support this study are available from the corresponding author upon reasonable request. The RNA-seq data in this study were deposited in the Gene Expression Omnibus (GEO) database under accession number GSE167233. The mass spectrometry proteomics data have been deposited to the ProteomeXchange Consortium via the jPOST repository with the dataset identifier PXD038094. Source data are provided with this paper.

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

## Acknowledgements
This work was supported in part by a Grant-in-Aid for Scientific Research on Innovative Areas and other grants from the Japan Society for the Promotion of Science (JSPS) (21H02692, 16H06574, and 15H04703 to M.T.; 15K19019 and 17K08650 to Y.K.), and by grants from JST CREST (JPMJCR2022) and the Princess Takamatsu Cancer Research Fund to M.T.

## Author contributions
S.O., Y.K., K.Y., Y.T., and J.N. conducted the experiments. S.O., Y.K., K.Y., and Y.F. analyzed the sequencing data. S.O., Y.K., K.Y., Y.T., H.K.H., M.O., Y.F., and M.T. designed the experiments and analyzed the data. S.O., Y.K., and M.T. wrote the paper.

## Competing interests
The authors declare no competing interests.
