## [Peer Review File · Nature Communications]

REVIEWER COMMENTS

Reviewer #1 (Remarks to the Author):

In the current study, Seina and colleagues identified ERK-dependent phosphorylation sites on NELF-A, a subunit of negative elongation factor (NELF). The authors demonstrated that ERK-mediated phosphorylation facilitates NELF eviction from the paused complex and promotes pause release and elongation. Furthermore, based on their results, authors suggested PP2A as a phosphatase of NELF-A; and the opposing actions of ERK and PP2A control the expression of IEGs, presumably through balancing the level of NELF-A phosphorylation. Finally, the authors pointed out the coexistence of ERK hyperactive oncogenic mutations and PP2A loss-of-function characteristics and their relevance in cancers.

The experiments were designed thoughtfully and executed accordingly. The paper is written well, and the data support the authors' conclusion in most cases. Thus, this study is very timely and will help to move the field forward. However, the following concerns are required to be addressed before publication in Nature Communications.

1. Authors can consider moving some redundant studies from the main Figure(s) to the Extended Data Figure(s). That will make it easier for the general audience to follow the story and minimize the figures' crowdedness. For example, Fig 1f and 1j can be relocated to the corresponding Extended Data Figure.
2. Fig. 1f: It looks that after an hour of ERK activation by either EGF or TPA, the slower migrating bands are disappearing. Does that correlate with prolonged ERK activation and downregulation of gene expression after an hour of ERK activation?
3. A seemingly puzzling observation in Fig 1j—knocking down Cdk9 causes the reduction in Pol II CTD Ser2 phosphorylation, leading to blockage in pause release; however, authors still can see activated ERK-mediated hyperphosphorylation of NELF-A, a pause releasing event, as per the authors' proposed model. To end this confusion, it is recommended to measure gene expression by RT-qPCR and Pol II distribution by ChIP after inhibiting or depleting Cdk9 in the context of HEK293-WT or -4A cells.
4. The part of establishing PP2A as a phosphatase of NELF-A is weak, as the authors used various inhibitors (Fig. 5b-d), which have off-targets; a potential one is PP1. Moreover, the authors claimed Na3VO4 did not stabilize NELF-A phosphorylation; however, visually, it looks it has the same effects as okadaic acid (Fig. 5b). Furthermore, the *in vitro* phosphatase assay (Fig. 5h) does not imply specificity of PP2A towards phospho-NELF-A. Additionally, the gene expression studies using okadaic acid (Fig. 5k & l) do not suggest the specific effects of PPA inhibition, as this drug can hit PP1 at a much lower concentration than the dose authors used in this study.

Therefore, to establish PP2A as a phosphatase of NELF-A and possible roles in controlling the expression of IGFs through regulating the NELF-A phosphorylation, this reviewer recommends using a comparatively cleaner system based on the recent reports—recruitment of PP2A to the elongation complex in association with Integrator complex to control Spt5 and Pol II phosphorylation at various positions to regulate the pause release and elongation (Huang et al., 2020, *Molecular Cell* 80, 1–14; Zheng et al., 2020, *Science* 370, 1059; Vervoort et al., 2021, *Cell* 184, 3143–3162). Authors can deplete Integrator subunits, Ints6 or Ints8, by RNAi approach and study the NELF-A phosphorylation, Pol II distribution, and gene expression to document the specificity of PP2A activity towards NELF-A.

5. On the same line, in Fig. 6h, the induction of IEGs' expression upon overexpression of a PP2A inhibitor can be a consequence of releasing barriers from the phosphorylation of Pol II and Spt5 CTD, as suggested by the recent reports (Huang et al., 2020, *Molecular Cell* 80, 1–14; Zheng et al., 2020, *Science* 370, 1059; Vervoort et al., 2021, *Cell* 184, 3143–3162).

Reviewer #2 (Remarks to the Author):

This is a very nice work aimed to understand the role of ERK-mediated phosphorylation of NELF in transcriptional control and tumorigenesis. The authors initially identified NELF-A, the largest subunit of the negative elongation factor NELF, as a novel substrate for ERK by using a yeast two-hybrid screen and clearly showed that this phosphorylation is involved in ERK-induced expression of IEGs by affecting Pol II pause release. Moreover, the authors identified PP2A as the phosphatase counteracting the ERK-mediated phosphorylation and presented evidence indicating that PP2A inhibitor proteins such as SET and CIP2A are involved in IEG expression and cancer progression. Over all, this study is well designed, and the results are clear. I have listed specific comments and questions below that could be used to strengthen the conclusions of this study.

(1) Fig. 3a and Fig. 4a: Residual amounts of endogenous NELF-A and overexpression levels of exogenous NELF-A should be quantified. This is crucial because phenotypic differences between WT and 4A are not so large.

(2) I could not look up the RNA-seq data on NCBI GEO, because GSE167233 is currently private.

(3) Of the 210 EGF-inducible genes identified, only 46 genes were significantly attenuated in the 4A cells. How could this happen? I think this issue is worth discussing.

(4) Considering that PP2A plays a multifaceted role in regulating Pol II transcription, including as a structural component of the Integrator–PP2A complex INTAC, it is hard to conclude from the current data that the primary point of action of PP2A inhibitors is NELF. In Fig. 5k, for example, JUNB and FOSB mRNA levels were upregulated by OA only in NELF-A WT cells but also in NELF-A 4A cells. There are two possible experiments that can be conducted to address this point. First, cells harboring NELF-A with acidic amino acid substitutions (4D or 4E) can be compared with WT and 4A cells. Second, EGF-inducible genes that behave similarly in WT and 4A cells (Fig. 3e) may be used to examine responses to ocadaic acid in Fig. 5k. The latter idea is also applicable to the experiments shown in Fig. 5i. Since the first option is easy to say but hard to do, I think that the second one is a viable option.

Reviewer #3 (Remarks to the Author):

RNAPII promoter proximal pausing and subsequent release into elongation is key to mRNA synthesis of many genes and is tightly regulated by various negative and positive factors. In an effort to understand the regulation of immediate early gene expression by growth factors, Ohe and colleagues identified NELF-A, a negative transcription elongation factor, as a novel substrate of ERK. Phosphorylation of NELF-A by ERK at multiple sites facilitated the removal of NELF-A from the promoters of certain immediate early genes, which are required for their expression and growth factor-mediated cell proliferation. They also found that cancer cells with constitutively active ERK triggered the NELF-A phosphorylation and

immediate early gene expression. Furthermore, they demonstrate that the effect of ERK on NELF-A was counter balanced by PP2A dephosphorylation mediated by PP2A inhibitor proteins SET and CIP2A.

While the biochemical data identifying the NELF-A as a substrate for ERK and its involvement in the immediate early gene expression were convincing, data for ERK-mediated NELF-A phosphorylation and its contribution to cancer cell proliferation were not sufficient. In addition, the conclusion that NELF-A is not phosphorylated by CDK9 is contradictory to previous studies. It is not clear how the cells selectively or sequentially employ ERK and CDK9 for the removal of NELF-A for transcription elongation.

Specific comments:

Fig.1J: the authors used siRNA to knockdown the expression of CDK9 and concluded that CDK9 was not required for NELF-A phosphorylation. However, the remaining amount of CDK9 after KD could also explain the unchanged elevated phosphorylation of NELF-A. Under normal conditions, large portions of P-TEFb are sequestered in an inactive state, upon siRNA KD, the remaining P-TEFb could still be sufficient to phosphorylate NELF-A. The authors should use CDK9 inhibitors to exclude the possible involvement of CDK9. Could the authors knock down the expression of ERK and examine the phosphorylation status of NELF-A?

Fig.2B&2C: S363A mutant showed clear different phosphorylation levels between MEK1-induced and TPA-induced phosphorylation. What might cause this discrepancy considering both experiments were done in 293T cells? Why did the authors use two different approaches to measure the phosphorylated NELF-A in these experiments (Phospho-tag vs. Pro-Q)?

Fig. 3I and K: the phosphorylated RNAPII should be included in these experiments. ChIP experiments in ERK KD or ERK inhibitor-treated cells should also be included.

Fig. 4b: Why the authors switch to the P-SP/TP antibodies for the detection of phosphorylated form of NELF-A? The authors used several different methods to probe the phosphorylated NELF-A. Is there any reason that they keep switching these methods?

Fig. 4: HaCaT is an immortalized nontumorigenic human epidermal cell line. It is very rare to use this cell line to form tumors. Although the authors referred two papers for using this cell line for tumor formation, none of them actually showed these cells could directly form tumors in nude mice (REF. 27&28). The authors should use one of the cancer cell lines in Fig. 5 for these cancer related experiments.

Fig. 5H: the NELF-A-4A mutant should be included as control. The authors suggested that "oncogene-induced, constitutive ERK activity alone is insufficient to provoke aberrant NELF-A phosphorylation and IEG expression in cancer cells". Does this apply to growth factor-induced expression of IEG in 293T or HeLa cells?

Fig. 6: The correlation of the SET and CIP2A with IEG expression is interesting. The significance of this study would be strengthened if the authors could demonstrate the phosphorylation of NELF-A and the expression of IEG in human cancer samples, especially those samples with constitutively active ERK. Overexpression of SET or CIP2A in cancer cell lines as in Fig. 6H is not sufficient.

Point-by-point responses to referees' comments

First, we deeply thank all the reviewers for their positive and insightful comments and suggestions on the manuscript.

Reviewer #1:

1. Authors can consider moving some redundant studies from the main Figure(s) to the Extended Data Figure(s). That will make it easier for the general audience to follow the story and minimize the figures' crowdedness. For example, Fig 1f and 1j can be relocated to the corresponding Extended Data Figure.

In accordance with the reviewer's advice, we moved the old Fig. 1i to the Supplementary Fig. 1 (as new Supplementary Fig. 1c). As for the original Fig. 1f, we retain this in the main Fig. 1, because we believe that Fig. 1f provides important information concerning the relationship between the intensity of ERK activity and the phosphorylation status of NELF-A. The figure legends and text have been changed accordingly.

2. Fig. 1f: It looks that after an hour of ERK activation by either EGF or TPA, the slower migrating bands are disappearing. Does that correlate with prolonged ERK activation and downregulation of gene expression after an hour of ERK activation?

As the reviewer pointed out, in Fig. 1f, we showed that, when HeLa cells were stimulated with EGF (but not TPA), the slower migrating (phosphorylated) NELF-A bands disappeared at (60 or) 120 min after EGF addition. Therefore, we monitored the mRNA expression levels of representative IEGs by qRT-PCR at 0 (before stimulation), 30 (when ERK is strongly activated and therefore NELF-A is highly phosphorylated), and 120 min (when ERK is still moderately activated, but NELF-A is almost completely dephosphorylated) after EGF addition. As shown in the new Supplementary Fig 1b, the expression levels of these IEGs are significantly reduced at 120 min after EGF stimulation. Thus, consistent with our conclusion, NELF-A dephosphorylation correlates with downregulation of the IEGs.

Regarding the correlation between NELF-A dephosphorylation and ERK activity, we had shown that following mitogenic stimulation, NELF-A phosphorylation gradually declined and returned to a basal level by about 2 h, even though moderate ERK activity persisted for longer time periods (Fig. 1f, and Fig. 5b-d, and g). This is because, as we demonstrated in this manuscript, the phosphorylation state of NELF-A is principally determined by the balance between ERK and PP2A activities, and its phosphorylation requires a relatively high level of ERK activity to overcome PP2A-mediated dephosphorylation. This is discussed in page 17, lines 12-25).

3. A seemingly puzzling observation in Fig 1j (now Fig 1i)—knocking down Cdk9 causes the reduction in Pol II CTD Ser2 phosphorylation, leading to blockage in pause release; however, authors still can see activated ERK-mediated hyperphosphorylation of NELF-A, a pause releasing event, as per the authors' proposed model. To end this confusion, it is recommended to measure gene

expression by RT-qPCR and Pol II distribution by ChIP after inhibiting or depleting Cdk9 in the context of HEK293-WT or -4A cells.

I think this comment may be due to some misunderstanding. In this study, we do not deny the importance of the P-TEF-b (CDK9)-mediated Pol II CTD Ser2 phosphorylation for the induction of Pol II pause release, nor do we claim that ERK-mediated NELF-A phosphorylation alone is sufficient for the induction of Pol II pause release. Instead, we argue that both ERK-mediated NELF-A phosphorylation and P-TEF-b-mediated Pol II CTD Ser2 phosphorylation are critical for the robust induction of Pol II pause release and the resulting productive Pol II elongation on IEGs (please see page 16, lines 26 and 27, and page 17, lines 1-7). In particular, in the old Fig. 1j (now Fig. 1i), we demonstrated that growth factor-induced NELF-A phosphorylation is mediated by ERK but not by P-TEF-b, while Pol II CTD Ser2 phosphorylation is mediated by P-TEF-b but not by ERK. These two phosphorylation events synergistically promote growth factor-induced eviction of NELF from paused Pol-II and consequent productive Pol-II elongation on IEGs.

4. The part of establishing PP2A as a phosphatase of NELF-A is weak, as the authors used various inhibitors (Fig. 5b-d), which have off-targets; a potential one is PP1. Moreover, the authors claimed Na3VO4 did not stabilize NELF-A phosphorylation; however, visually, it looks it has the same effects as okadaic acid (Fig. 5b). Furthermore, the in vitro phosphatase assay (Fig. 5h) does not imply specificity of PP2A towards phospho-NELF-A. Additionally, the gene expression studies using okadaic acid (Fig. 5k & l) do not suggest the specific effects of PPA inhibition, as this drug can hit PP1 at a much lower concentration than the dose authors used in this study.

Therefore, to establish PP2A as a phosphatase of NELF-A and possible roles in controlling the expression of IGFs (*note: I think this must be IEGs*) through regulating the NELF-A phosphorylation, this reviewer recommends using a comparatively cleaner system based on the recent reports—recruitment of PP2A to the elongation complex in association with Integrator complex to control Spt5 and Pol II phosphorylation at various positions to regulate the pause release and elongation (Huang et al., 2020, Molecular Cell 80, 1–14; Zheng et al., 2020, Science 370, 1059; Vervoort et al., 2021, Cell 184, 3143–3162). Authors can deplete Integrator subunits, Ints6 or Ints8, by RNAi approach and study the NELF-A phosphorylation, Pol II distribution, and gene expression to document the specificity of PP2A activity towards NELF-A.

In order to address these issues, we conducted the following experiments.

First, concerning the experiments shown in Figure 5, we repeated some key experiments using the highly specific PP2A inhibitors (rubratoxin A and cytostatin), which do not inhibit PP1 (PMID: 20028386, cited), instead of okadaic acid (OA), and the data have been added as new supplementary Fig. 5d, e, g, h. In the supplementary Fig. 5d and e, we stimulated HEK293-WT and -4A cells with EGF, in the presence or absence of rubratoxin A or cytostatin, and quantified the expression levels of the representative IEGs (i.e., *JUNB* and *FOSB*) by qRT-PCR. Consistent with the results obtained with OA (Fig. 5k), in the presence of the highly specific PP2A inhibitors, the levels of EGF-induced *JUNB/FOSB* expression were markedly enhanced in HEK293-WT cells, but only weakly in HEK293-4A cells. Moreover, in the supplementary Fig. 5g and h, we confirmed that, similar to the results obtained with OA (Fig. 5l), treatment

of H1299 cancer cells with rubratoxin A or cytosatin markedly upregulated the expression levels of *JUNB/FOSB*. Thus, the highly specific PP2A inhibitors, cytosatin and rubratoxin A, showed essentially identical effects to those produced by OA. These findings further bolster our conclusion that PP2A counteracts NELF-A phosphorylation, thereby suppressing ERK-mediated IEG expression.

Second, in accordance with the reviewer's request, we conducted the suggested experiment (i.e., depletion of INTS6 by siRNA). However, we must note that:

1) Although, as the reviewer mentioned, previous studies reported that the Integrator complex (INTAC) (a non-canonical PP2A complex containing Integrator, PP2A-A, and PP2A-C subunits, but lacking PP2A-B subunit) dephosphorylate RNA-Pol II CTD and Spt5, there has been no evidence that INTAC also dephosphorylates NELF-A. Therefore, there is no compelling reason to believe that inhibition of INTAC affects NELF-A phosphorylation.

2) Moreover, we demonstrated in the present study that expression of CIP2A significantly enhanced NELF-A phosphorylation by inhibiting PP2A (Fig. 5f and 6g, h). Importantly, it has been known that CIP2A selectively inhibits only canonical (PP2A-B subunit-containing) PP2A complex by directly interacting with the PP2A-B subunit (PMID: 28174209 and 28884018). Therefore, our findings strongly suggest that NELF-A is mainly dephosphorylated by canonical PP2A, but not by non-canonical (PP2A-B subunit-lacking) INTAC.

Indeed, as shown in the new Supplementary Fig. 5a and b, depletion of INST6 (a critical component of INTAC) by two different siRNAs can moderately enhanced EGF-induced Pol II CTD (Ser2) phosphorylation (Supplementary Fig. 5a), but it did not affect the phosphorylation status of NELF-A (Supplementary Fig. 5b). These findings indicate that the canonical PP2A, but not the non-canonical INTAC, is the major phosphatase that dephosphorylates NELF-A in cells. In any case, this issue (i.e., whether INTAC is also involved in NELF-A dephosphorylation or not) is outside our current scope.

5. On the same line, in Fig. 6h, the induction of IEGs' expression upon overexpression of a PP2A inhibitor can be a consequence of releasing barriers from the phosphorylation of Pol II CTD and Spt5, as suggested by the recent reports (Huang et al., 2020, *Molecular Cell* 80, 1–14; Zheng et al., 2020, *Science* 370, 1059; Vervoort et al., 2021, *Cell* 184, 3143–3162).

As we mentioned above, since CIP2A selectively inhibits canonical (PP2A-B subunit-containing) PP2A by directly interacting with the PP2A-B subunit, it is unlikely that expressed CIP2A binds to and inhibits the non-canonical (PP2A-B subunit-lacking) INTAC complex to enhance the phosphorylation of Pol II CTD and Spt5 in Fig. 6h. Furthermore, using the NELF-A(4A) mutant, we confirmed that the upregulation of IEGs that is induced by PP2A inhibition is mainly mediated by NELF-A phosphorylation (Fig. 5k and the new Supplementary Fig. 5d, e), and that PP2A can directly dephosphorylate NELF-A in vitro (Fig. 5h). We believe that these and other data presented in this manuscript sufficiently support our conclusion.

Reviewer #2:

1) Fig. 3a and Fig. 4a: Residual amounts of endogenous NELF-A and overexpression levels of exogenous NELF-A should be quantified. This is crucial because phenotypic differences between WT and 4A are not so large.

In accordance with the reviewer's request, we quantified the expression levels of endogenous and exogenous NELF-A proteins by densitometry analyses of the Western blots in Fig. 3a and Fig. 4a, and the data are shown as new Supplementary Fig. 3a (for the HEK293-derived cell lines) or are incorporated into Fig. 4a (for the HaCaT-derived cell lines). The results showed that more than 90% of endogenous NELF-A was depleted by shRNA-mediated knockdown, and the expression levels of Myc-NELF-A proteins (WT and 4A) were comparable in the HEK293-derived cell lines (Supplementary Fig. 3a) and in the HaCaT-derived cell lines (Fig. 4a).

2) I could not look up the RNA-seq data on NCBI GEO, because GSE167233 is currently private.

The password (token) for the GSE167233 data is "gnkdmywaxbgnfsb". Please use this password to access the data.

3) Of the 210 EGF-inducible genes identified, only 46 genes were significantly attenuated in the 4A cells. How could this happen? I think this issue is worth discussing.

We thank the reviewer for this thoughtful comment. As we described in the Introduction section, previous studies showed that not all genes are necessarily regulated by promoter-proximal pausing (PPP) of RNA Pol II. For instance, it has been reported that PPP is seen in approximately 60% of expressed mammalian genes (PMID: 27259512, cited). Furthermore, many lines of evidence have shown that the levels of PPP of Pol II (termed "Pausing Index") vary among individual paused genes, because they can be readily influenced by chromatin structure and local sequence composition near the transcription start site on individual genes (PMID: 27259512; 25773599; 21460038; 27353326). Therefore, in our experiment, EGF-inducible genes with relatively high Pausing Indices are likely to be preferentially detected as genes whose expression levels are significantly attenuated in HEK293-4A cells, compared with HEK293-WT cells. We briefly mentioned this point in Discussion (page 17, line 11).

4) Considering that PP2A plays a multifaceted role in regulating Pol II transcription, including as a structural component of the Integrator-PP2A complex INTAC, it is hard to conclude from the current data that the primary point of action of PP2A inhibitors is NELF. In Fig. 5k, for example, JUNB and FOSB mRNA levels were upregulated by OA only in NELF-A WT cells but also in NELF-A 4A cells. There are two possible experiments that can be conducted to address this point. First, cells harboring NELF-A with acidic amino acid substitutions (4D or 4E) can be compared with WT and 4A cells. Second, EGF-inducible genes that behave similarly in WT and 4A cells (Fig. 3e) may be used to examine responses to okadaic acid in Fig. 5k. The latter idea is also applicable to the experiments shown in Fig. 5i (note: This must be Fig. 5l). Since the first option is easy to say but hard to do, I think that the second one is a viable option.

We thank the reviewer for this insightful comment. In accordance with the reviewer's advice, we conducted the second experiment: We examined the responses of two EGF-inducible genes that behave similarly in HEK293-WT and -4A cells (i.e., *BTG1* and *SOWAHC*) to okadaic acid (OA), and the data have been added as new Supplementary Fig. 5c and 5f.

In the new supplementary Fig. 5c, we treated HEK293-WT and -4A cells with EGF in the presence or absence of OA, and quantified the expression levels of *BTG1* and *SOWAHC* by qRT-PCR. The results demonstrated that, unlike *JUNB* and *FOSB*, OA treatment did not significantly affect EGF-induced expression of *BTG1* or *SOWAHC*. Thus, these findings further corroborate our conclusion that PP2A counteracts growth factor-induced NELF-A phosphorylation, thereby suppressing the induction of IEGs whose expression is mainly regulated by Pol II pausing and its release (such as *JUNB* and *FOSB*, but not *BTG1* or *SOWAHC*). Furthermore, in the new supplementary Fig. 5f, we treated A375 cancer cells with OA and monitored the expression levels of *BTG1* and *SOWAHC*. Again, OA treatment did not significantly affect the expression levels of *BTG1* and *SOWAHC*. These findings further support our original conclusion.

Reviewer #3:

1) Fig.1J (now Fig 1i): the authors used siRNA to knockdown the expression of CDK9 and concluded that CDK9 was not required for NELF-A phosphorylation. However, the remaining amount of CDK9 after KD could also explain the unchanged elevated phosphorylation of NELF-A. Under normal conditions, large portions of P-TEFb are sequestered in an inactive state, upon siRNA KD, the remaining P-TEFb could still be sufficient to phosphorylate NELF-A. The authors should use CDK9 inhibitors to exclude the possible involvement of CDK9. Could the authors knock down the expression of ERK and examine the phosphorylation status of NELF-A?

In accordance with the reviewer's request, we conducted the suggested experiment, and the data have been added as new Supplementary Fig. 1d. In this figure, we stimulated HeLa cells with TPA in the presence or absence of a CDK9 inhibitor (DRB), and the phosphorylation status of NELF-A were assessed by Phos-tag SDS-PAGE. The result demonstrated that, although treatment of cells with DRB abolished CDK9-mediated phosphorylation of Pol II CTD(Ser2), it did not inhibit TPA-induced NELF-A phosphorylation at all. These findings are consistent with those obtained by siRNA-mediated CDK9 knockdown (Fig 1i) and with recent studies showing that NELF dissociation can occur independently of CDK9 (PMID: 25263592, 29523821, and 32385332, cited). Thus, this result further corroborates our conclusion that mitogen-induced NELF-A phosphorylation is primarily mediated by ERK but not by CDK9.

2) Fig.2B&2C: S363A mutant showed clear different phosphorylation levels between MEK1-induced and TPA-induced phosphorylation. What might cause this discrepancy considering both experiments were done in 293T cells? Why did the authors use two different approaches to measure the phosphorylated NELF-A in these experiments (Phospho-tag vs. Pro-Q)?

Phos-tag SDS-PAGE technique can visualize phosphorylated proteins as up-shifted bands, and is thus useful for the easy detection of phosphorylated proteins. However, one technical issue of this approach is that the degree of the band shift in Phos-tag SDS-PAGE does not always simply reflect the number of phosphorylated sites on a protein, but rather, it is sometimes affected by experimental conditions used [including amino acid sequence of a protein of interest, and composition of lysis buffer and running buffer (e.g., salt content and concentration, pH, etc.)]. In the case of Phos-tag SDS-PAGE analysis of NELF-A protein, as shown in Fig. 2b, while its Ser 363 phosphorylation led to a pronounced retardation in electrophoretic mobility of NELF-A, the phosphorylation at other 3 sites (S393/T396/T399) led to relatively moderate band-shifts. Therefore, we utilized 3 different approaches (Phos-tag SDS-PAGE, Pro-Q Diamond phosphoprotein staining, and Western blot analysis using an anti-phospho-SP/TP monoclonal antibody) to rigorously evaluate NELF-A phosphorylation, and confirmed that virtually identical results were obtained with all three approaches.

3)Fig. 3I and K: the phosphorylated RNAPII should be included in these experiments. ChIP experiments in ERK KD or ERK inhibitor-treated cells should also be included.

Regarding the phosphorylation states of RNA pol II, we had clearly demonstrated in the original Supplementary Fig 3b (now Supplementary Fig. 3c) that there were no significant differences between

HEK293-WT and -4A cells in the phosphorylation levels of RNA Pol II-CTD at Ser2 and Ser5. In addition, our ChIP analyses using anti-RNA Pol II antibody showed the statistically significant, reliable results in Fig. 3i, k and Supplementary Fig. 3 f, g (these experiments were repeated at least three times with similar results). Furthermore, in many previous studies (PMID: 16880520, 32385332, 30150253), anti-RNA Pol II antibodies have been used for ChIP analysis of RNA Pol II pausing and release. Thus, we believe that there is no compelling reason to perform the ChIP analyses using a particular phospho-specific RNA Pol II antibody, instead of an anti-RNA Pol II antibody.

As for the ChIP experiments using ERK inhibitor-treated cells, we think that the suggested experiment would not provide conclusive results regarding ERK-mediated Pol II pause release. As we described in the main text, ERK not only phosphorylates NELF-A but also phosphorylates and activates several transcription factors (e.g., ELK-1 and Sp1) that control the transcription initiation process of IEGs. Therefore, suppression of ERK activity in cells inevitably inhibits not only NELF-A phosphorylation-mediated Pol II pause release but also the transcription initiation step of IEGs including *FOS* and *JUNB* (i.e., It also inhibits the recruitment of Pol II and NELF to the transcription start sites of these genes). Thus, suppression of ERK activity in cells will compromise all the critical steps in the transcription cycle of IEGs. This will make it difficult to interpret the results obtained from the suggested experiment.

4) Fig. 4b: Why the authors switch to the P-SP/TP antibodies for the detection of phosphorylated form of NELF-A? The authors used several different methods to probe the phosphorylated NELF-A. Is there any reason that they keep switching these methods?

This is a similar comment to the comment 2) of this reviewer. Please see our response to the comment 2). Since immunoprecipitation (IP)-Western blot analysis is the technically easiest and least expensive method among the above-mentioned 3 approaches, we utilized IP-Western blot analysis using the P-SP/TP antibody in Fig. 4b.

5) Fig. 4: HaCaT is an immortalized nontumorigenic human epidermal cell line. It is very rare to use this cell line to form tumors. Although the authors referred two papers for using this cell line for tumor formation, none of them actually showed these cells could directly form tumors in nude mice (REF. 27&28). The authors should use one of the cancer cell lines in Fig. 5 for these cancer related experiments.

In order to further assess the effect of ERK-mediated NELF-A phosphorylation on in vivo tumorigenesis, we conducted the suggested experiment using H-Ras(G12V)-transformed HaCaT cells. Since this Ras-transformed HaCaT system has been widely used as an in vivo carcinogenesis model (PMID: 20101224, 7505755, 1714343, 9833775, 11583982, 22414291, etc.), we believe that this system would be more suitable than the use of a particular cancer cell line for precisely evaluating the role of NELF-A phosphorylation in the process of carcinogenesis in vivo. The data have been added as the new Fig. 4i, j and new supplementary Fig. 4. In these figures, we established HaCaT-WT and HaCaT-4A cells that stably express oncogenic H-Ras(G12V) mutant [termed Ras-HaCaT-WT and Ras-HaCaT-4A cells, respectively] (new Supplementary Fig. 4). These two cell lines were subcutaneously injected into nude mice, and tumor

growth was then monitored for 35 days. As shown in Fig. 4i, these H-Ras(G12V)-transformed cell lines developed aggressive tumors in nude mice, but the tumors formed by Ras-HaCaT-4A cells were significantly smaller than those derived from Ras-HaCaT-WT cells. Corresponding changes in tumor weight were also observed after sacrifice of the mice (Fig. 4j). Thus, these data further corroborate our conclusion that ERK-mediated NELF-A phosphorylation promotes tumor development and progression in vivo.

6) A) Fig. 5H: the NELF-A-4A mutant should be included as control. B) The authors suggested that “oncogene-induced, constitutive ERK activity alone is insufficient to provoke aberrant NELF-A phosphorylation and IEG expression in cancer cells”. Does this apply to growth factor-induced expression of IEG in 293T or HeLa cells?

A) In accordance with the reviewer’s request, we conducted the suggested experiment using NELF-A(4A) as control, and have replaced the original Fig. 5h with the new analysis. The revised results further support our original conclusion that PP2A directly dephosphorylates NELF-A.

B) As for the correlation of NELF-A phosphorylation status with ERK activity and IEG expression, we had shown that, following mitogenic stimulation, NELF-A phosphorylation gradually declined and returned to a basal level by about 2 h even though moderate ERK activity persisted for longer time periods in HeLa and HEK293 cells (Fig. 1f and Fig. 5d, g), and that this relatively rapid dephosphorylation of NELF-A is mediated by PP2A (Fig. 5b-h). Furthermore, we confirmed that, consistent with the relatively rapid dephosphorylation of NELF-A (Fig. 1f), the expression levels of the representative IEGs (*FOS*, *EGR1*, and *EGR2*) were significantly reduced at 120 min following EGF addition (new Supplementary Fig. 1b). We believe that these and other data presented in this manuscript sufficiently support our conclusion that NELF-A phosphorylation and the resulting IEG induction require a relatively high level of ERK activity to overcome PP2A-mediated dephosphorylation.

7) Fig. 6: The correlation of the SET and CIP2A with IEG expression is interesting. The significance of this study would be strengthened if the authors could demonstrate the phosphorylation of NELF-A and the expression of IEG in human cancer samples, especially those samples with constitutively active ERK. Overexpression of SET or CIP2A in cancer cell lines as in Fig. 6H is not sufficient.

We appreciate this thoughtful comment. However, analyses of NELF-A phosphorylation and IEG expression in human clinical cancer samples with constitutively active ERK and SET/CIP2A overexpression, while we do wish to study those in the future, will be beyond our current scope. We believe that the identification of NELF-A as a novel ERK substrate, and of its role in growth factor-induced transcription elongation of IEGs and in tumorigenesis is sufficiently important to be reported in *Nature Communications*.

REVIEWERS' COMMENTS

Reviewer #1 (Remarks to the Author):

The authors have addressed all of my concerns. Therefore, the manuscript can be accepted in its current format.

Reviewer #2 (Remarks to the Author):

I believe that all the issues raised by the reviewers, including myself, have been adequately addressed in the revised manuscript, along with the attached point-by-point responses.

Reviewer #3 (Remarks to the Author):

In this revised manuscript, the authors have addressed most of my concerns with additional experiments. However, there is still one concerns need to be addressed. In the new experiments with the Ras-HaCaT-WT and Ras-HaCaT-4A cells, the phosphorylation status of ERK, NELF-A and IEG expression (e.g., JUNB and FOSB) need to be examined. Without these data, it is not clear whether the phenotypes from these cells are indeed due to the activation of ERK and the phosphorylation of NELF-A. What is the status of those phosphatases in these H-RA expressing cells since H1299 cells with the NRASQ61K mutation display constitutively activated ERK without the up-regulation of IEGs.

Point-by-point response to referees' comments

We deeply thank all of the reviewers for their positive and insightful comments and suggestions on the manuscript.

Reviewer #1:

The authors have addressed all of my concerns. Therefore, the manuscript can be accepted in its current format.

Reviewer #2:

I believe that all the issues raised by the reviewers, including myself, have been adequately addressed in the revised manuscript, along with the attached point-by-point responses.

Response:

As shown above, the reviewers #1 and #2 have no further comments.

Referee #3:

In this revised manuscript, the authors have addressed most of my concerns with additional experiments. However, there is still one concerns need to be addressed. In the new experiments with the Ras-HaCaT-WT and Ras-HaCaT-4A cells, the phosphorylation status of ERK, NELF-A and IEG expression (e.g., JUNB and FOSB) need to be examined. Without these data, it is not clear whether the phenotypes from these cells are indeed due to the activation of ERK and the phosphorylation of NELF-A. What is the status of those phosphatases in these H-RA expressing cells since H1299 cells with the NRASQ61K mutation display constitutively activated ERK without the up-regulation of IEGs.

Response:

In accordance with the reviewer's request, we conducted the suggested experiments and the data have been added as new Supplementary Figs. 4b and 4c.

In these figures, we confirmed that, at steady-state (in the absence of EGF stimulation), no phosphorylation of NELF-A was detected in both Ras-HaCaT-WT and Ras-HaCaT-4A cells, even though these cell lines exhibited a moderately elevated basal ERK activity due to the stable expression of H-Ras(G12V) (Supplementary Fig. 4b), and, therefore, both cell lines exhibited low IEG expression at steady-state (Supplementary Fig. 4c). Thus, these findings further support our conclusion that PP2A counteracts NELF-A phosphorylation induced by ERK-activating oncogenes, and represses IEG expression.

In contrast, when these cells were stimulated with a growth factor (EGF), EGF-induced NELF-A phosphorylation (Supplementary Fig. 4b) and the expression of the representative IEGs (Supplementary Fig. 4c) were significantly repressed in Ras-HaCaT-4A cells as compared with Ras-HaCaT-WT. These results further corroborate our conclusion that ERK-mediated NELF-A

phosphorylation is crucial for growth factor-induced robust expression of IEGs, and explain why the tumors formed by Ras-HaCaT-4A cells were significantly smaller than those derived from Ras-HaCaT-WT in the mouse xenograft experiments shown in Fig. 5i and j.